# Zilebesiran as an Innovative siRNA-Based Therapeutic Approach for Hypertension: Emerging Perspectives in Cardiovascular Medicine

**DOI:** 10.3390/ijms262110717

**Published:** 2025-11-04

**Authors:** Petruta A. Morosan, Amelian M. Bobu, Alexandru Carauleanu, Radu Popa, Claudia F. Costea, Cristiana Filip, Catalin M. Buzduga, Emilia Patrascanu, Andrei I. Cucu, Razvan I. Tudosa, Roxana Covali, Anca Haisan

**Affiliations:** 1Faculty of Medicine, Grigore T. Popa University of Medicine and Pharmacy Iasi, 700115 Iasi, Romania; anca.morosan@umfiasi.ro (P.A.M.); ale.carauleanu@umfiasi.ro (A.C.); radu.popa@umfiasi.ro (R.P.); claudia.costea@umfiasi.ro (C.F.C.); cristiana.filip@umfiasi.ro (C.F.); catalin.buzduga@umfiasi.ro (C.M.B.); patrascanu.emilia@umfiasi.ro (E.P.); ana.covali@umfiasi.ro (R.C.); anca.haisan@umfiasi.ro (A.H.); 2Faculty of Medicine and Biological Sciences, Stefan cel Mare University of Suceava, 720229 Suceava, Romania; andrei.cucu@usm.ro; 3“St Maria” Emergency Children Hospital, 700309 Iasi, Romania; razvan-ionut.tudosa@email.umfiasi.ro

**Keywords:** zilebesiran, siRNA therapy, angiotensinogen, blood pressure, subcutaneously administered, RNA drug

## Abstract

Zilebesiran represents an innovative antihypertensive therapy employing small interfering RNA (siRNA) to inhibit hepatic angiotensinogen, a key regulator of the renin–angiotensin–aldosterone system. By directly targeting the source of angiotensin II production, zilebesiran offers a novel mechanism distinct from conventional antihypertensive treatments. In the clinical studies KARDIA-1 and KARDIA-2, zilebesiran demonstrated clinically significant reductions in systolic blood pressure, with effects lasting up to 24 weeks after a single subcutaneous injection. In KARDIA-1, doses of 300 mg and 600 mg administered every 6 months resulted in reductions of over 15 mmHg in systolic blood pressure at 3 months compared with placebo. KARDIA-2 further showed an additional reduction of up to 12.1 mmHg at 3 months when zilebesiran was used as an adjunct to standard antihypertensive therapy. KARDIA-3 is currently evaluating the therapy in a larger global population to assess its impact on major cardiovascular outcomes. Zilebesiran has demonstrated a favorable safety profile with minimal adverse events, offering potential advantages for patients with resistant or uncontrolled hypertension and those at high cardiovascular risk, especially where adherence to daily oral medications is challenging. Beyond blood pressure reduction, zilebesiran may protect target organs, including the heart, kidneys, and retina. In conclusion, zilebesiran represents a promising siRNA-based therapy that may redefine the management of difficult-to-control hypertension, offering durable, targeted, and patient-friendly treatment with broad cardiovascular benefits. Future studies will clarify its long-term safety, efficacy across diverse populations, and integration into personalized hypertension management strategies.

## 1. Introduction

Arterial hypertension is a highly prevalent condition worldwide, affecting over one billion individuals, and represents a major contributor to cardiovascular disease and premature mortality [1]. This disorder is characterized by a sustained elevation of blood pressure, often asymptomatic, which significantly increases the risk of stroke, heart disease, and renal failure. According to the 2024 European Society of Cardiology/European Society of Hypertension (ESC/ESH) Guidelines, arterial hypertension is defined as office blood pressure values of systolic ≥ 140 mmHg and/or diastolic ≥ 90 mmHg, confirmed by 24-h ambulatory blood pressure monitoring (≥130/80 mmHg) or home blood pressure monitoring (≥135/85 mmHg) [2,3]. In contrast, the 2025 American Heart Association (AHA) Hypertension Guidelines adopt slightly lower thresholds to facilitate earlier detection and timely intervention. Hypertension is classified based on repeated systolic and diastolic measurements as follows: elevated blood pressure is defined as systolic 120–129 mmHg and diastolic < 80 mmHg; stage 1 hypertension is defined as systolic 130–139 mmHg or diastolic 80–89 mmHg; and stage 2 hypertension is defined as systolic ≥ 140 mmHg or diastolic ≥ 90 mmHg [4].

According to data published by the World Health Organization (WHO) in 2023, an estimated 1.28 billion adults aged 30–79 years are living with arterial hypertension, with more than two-thirds residing in low- and middle-income countries [5]. However, 46% of affected individuals are unaware of their condition, and fewer than half (42%) receive appropriate diagnoses and treatments. Moreover, only 21% of patients with hypertension achieve optimal blood pressure control, highlighting substantial gaps in early detection, access to care, and treatment adherence [6].

Despite its high prevalence and considerable health burden, awareness of hypertension remains low: fewer than half of patients are aware of their condition, and less than 40% of the European population achieves adequate blood pressure control (target < 140/90 mmHg) while receiving antihypertensive therapy [6]. The Pan-African Society of Cardiology (PASCAR) has identified hypertension as the top priority for reducing cardiovascular disease burden on the continent, aiming to achieve a 25% control rate in Africa by 2025 [7]. Similarly, the World Hypertension League has set a target for Africa by 2030 to diagnose 80% of adults with hypertension, treat 80% of those diagnosed, and control 80% of those treated [8,9].

In the United States, the prevalence of hypertension has remained persistently high in recent years, with no significant decline. Among adults, prevalence was 32.8% in 2013–2014 and 32.0% in 2021–2023. Among individuals aged ≥75 years, a modest decline was observed, from 74.5% in 2017–2020 to 68.8% in 2021–2023 [10,11]. Concerning trends are also reported in Asia, where hypertension affects more than 245 million individuals over the age of 30 in South Asian countries, the majority of whom remain asymptomatic and unaware of their condition [12]. In China, approximately 270 million individuals are affected, yet only 13.8% achieve adequate blood pressure control, with limited access to treatment [13].

Given the significant global impact of hypertension on morbidity and mortality, a 33% reduction in prevalence by 2030 has been established as a key target within the Global Action Plan for the Prevention and Control of Non-Communicable Diseases [5,14]. This challenge is further compounded by difficulties in achieving sustained blood pressure control, which result not only from variability in pharmacological response but also from non-pharmacological factors such as poor adherence, unhealthy lifestyle behaviors, and drug-related side effects [15].

To improve treatment effectiveness, both European and American guidelines recommend combination therapy using multiple classes of antihypertensive agents, including diuretics, beta-blockers, calcium channel blockers, and renin–angiotensin–aldosterone system (RAAS) inhibitors, with the addition of mineralocorticoid receptor antagonists and alpha-receptor blockers in selected cases, in order to achieve optimal blood pressure control and reduce cardiovascular risk [2,3,4,16,17]. In clinical practice, some patients present with a form of difficult-to-control hypertension, defined as blood pressure remaining above target levels despite the concurrent use of at least three different classes of antihypertensive medications [18]. In real-world practice, approximately 54% of treatment regimens involve monotherapy, while 46% involve more than one subclass of antihypertensive agents, either as multiple pills (polytherapy, 34%) or as a single fixed-dose combination (FDC) pill (12%). ACE inhibitors (Angiotensin-Converting Enzyme inhibitors) and β-blockers are the most commonly prescribed medications. Trends from 2010 to 2019 show a decrease in the use of some monotherapy subclasses (ACE inhibitors: 19.1% to 15.4%; β-blockers: 16.2% to 10.8%; alpha blockers: 2.4% to 0.5%), while the use of ARBs (Angiotensin II Receptor Blockers) increased from 5.9% to 7.9%, and thiazide diuretics from 2.2% to 4.5% [4,19,20]. Overall, ACE inhibitors or ARBs were used in 59.2% and β-blockers in 43.8% of patients. These data highlight that despite widespread use of multiple drug classes, optimal blood pressure control remains suboptimal in many patients.

Furthermore, recent genetic research, particularly genome-wide association studies (GWAS), has identified specific genomic regions linked to poor response to antihypertensive treatment, underscoring the critical role of genetic variability in the pathogenesis of this condition [21]. These findings support the concept that a personalized, genotype-guided therapeutic approach may optimize blood pressure control and improve clinical outcomes in resistant hypertension [22].

## 2. Methods

A comprehensive literature search was conducted in PubMed, Scopus, Web of Science, and ClinicalTrials.gov databases to identify relevant studies on zilebesiran and other small-interfering RNA (siRNA)-based therapies for hypertension. The search covered publications from January 2010 to October 2025. The following keywords and combinations were used: “zilebesiran”, “siRNA therapy”, “angiotensinogen inhibition”, “RNA interference”, “hypertension”, and “cardiovascular disease”.

Inclusion criteria comprised original research articles, clinical trials, and reviews written in English that reported preclinical or clinical findings on zilebesiran or related siRNA-based interventions in hypertension. Exclusion criteria included conference abstracts without full text, non-English publications, editorials, commentaries, and studies unrelated to cardiovascular applications.

### Pathophysiological Mechanisms of Hypertension

Essential hypertension is characterized by a complex pathophysiological background involving multiple neurohormonal, endothelial, and inflammatory dysfunctions that contribute to vascular remodeling and disturbances in fluid and electrolyte balance [23].

In recent years, novel classes of oral antihypertensive agents have emerged, including non-steroidal mineralocorticoid receptor antagonists, aldosterone synthase inhibitors, and dual endothelin receptor antagonists. These agents have demonstrated efficacy in lowering both systolic and diastolic blood pressure, as measured in-office and through 24-h ambulatory blood pressure monitoring [24]. Their mechanisms of action involve reducing sodium retention, limiting vascular injury, and modulating vasoconstriction, thereby offering additional therapeutic benefit, particularly in patients with uncontrolled or resistant hypertension [25,26].

More recently, RNA-based therapies targeting the renin–angiotensin–aldosterone system (RAAS) have shown promise as innovative strategies for long-term blood pressure control. These approaches allow for infrequent administration—potentially as little as twice per year—thus improving adherence and long-term outcomes [27]. Small interfering RNA (siRNA)–based injectable therapies, in particular, represent a major advancement. By targeting hepatic angiotensinogen, a critical component of the RAAS, these therapies can effectively modulate blood pressure regulation. For instance, zilebesiran, a long-acting siRNA therapeutic, has demonstrated the potential to substantially improve treatment adherence and achieve sustained blood pressure reduction [28,29].

## 3. From the Renin–Angiotensin System to RNA-Based Therapies: Innovative Approaches to Modulating Angiotensinogen Expression

The renin–angiotensin system (RAS) plays a central role in the regulation of blood pressure, and its inappropriate activation contributes to hypertension by influencing vascular tone, circulating volume, ionic balance, and aldosterone synthesis—processes that ultimately promote structural tissue alterations and target-organ damage. The key effector of the RAS is angiotensin II (Ang II), generated from angiotensin I (Ang I) through the action of angiotensin-converting enzyme (ACE). Ang I is, in turn, derived from angiotensinogen (AGT), a protein synthesized in the liver, upon cleavage by renin (Figure 1).

An innovative and promising therapeutic strategy for the control of hypertension involves the inhibition of hepatic angiotensinogen (AGT) synthesis, thereby reducing the formation of angiotensin I and II and consequently attenuating the activation of angiotensin II type 1 and type 2 receptors [30]. Two recent approaches have been developed to therapeutically inhibit AGT gene expression: antisense oligonucleotides (ASOs) and small interfering RNAs (siRNAs).

ASOs are short, single-stranded RNA molecules (8–50 nucleotides) designed to be complementary to a specific target mRNA. Upon entering the cell via endocytosis, they hybridize with the target mRNA either in the cytoplasm or the nucleus and recruit the enzyme RNase H, which cleaves and degrades the mRNA. This process ultimately prevents the translation of the target protein [31,32,33].

RNAi is an evolutionarily conserved endogenous mechanism by which short RNA fragments, such as siRNAs, specifically suppress gene expression. This process, discovered by Andrew Fire and Craig Mello and awarded the Nobel Prize in 2006, relies on the superior efficiency and specificity of double-stranded RNA compared to single-stranded RNA. The therapeutic exploitation of RNAi has since emerged as a promising avenue for targeted gene silencing [30,34,35].

Although the liver represents the main site of angiotensinogen (AGT) synthesis, recent studies have shown that AGT can also be produced, to a lesser extent, in extrahepatic tissues such as the brain, heart, kidneys, and adipose tissue. However, the contribution of these local sources to the total circulating AGT pool is minimal. Experimental models with >99% suppression of hepatic AGT expression demonstrated that extrahepatic synthesis is insufficient to sustain systemic angiotensin II (Ang II) production [36,37]. Nevertheless, it is now well recognized that an independent intrarenal renin–angiotensin system exists, capable of generating Ang II locally from intratubular angiotensinogen. This locally produced Ang II plays a crucial role in the regulation of sodium reabsorption in both proximal and distal nephron segments, thereby contributing to fluid balance and, under pathological conditions, to the development of angiotensin-dependent hypertension.

In response to the drastic reduction in AGT availability, the renin–angiotensin system activates a compensatory mechanism characterized by a marked increase in plasma renin levels. This compensatory renin surge accelerates the consumption of the residual AGT, paradoxically leading to complete substrate depletion and an almost total disappearance of Ang II [36]. Therefore, while extrahepatic tissues are capable of synthesizing AGT locally, their contribution cannot effectively compensate for a significant decrease in hepatic AGT production. The compensatory upregulation of renin release is limited in its capacity to restore Ang II levels, highlighting the liver’s essential role as the central regulator of systemic AGT synthesis and the homeostasis of Ang I and Ang II [38].

## 4. General Mechanism of siRNA Action

Small interfering RNAs (siRNAs) are short double-stranded RNA molecules (20–24 base pairs) characterized by two-nucleotide 3′ overhangs on each strand [39,40,41]. siRNAs are small double-stranded RNA molecules that suppress the expression of specific genes, thereby preventing the production of the corresponding protein in the cell [40,42]. In principle, any gene of interest can be targeted by siRNAs, provided that an appropriate complementary nucleotide sequence along the target mRNA is selected [43]. Furthermore, siRNAs have been shown to accumulate in hepatic endosomes, creating a depot effect with sustained release [25,44]. The major drawback of siRNA therapy is its potential to activate Toll-like receptor 3 (TLR3), thereby negatively affecting the hematologic and lymphatic systems [45,46]. The main steps in their mechanism of action are as follows:Cellular uptake: siRNAs enter the cell through endocytosis.Processing by Dicer: once inside the cytoplasm, siRNAs are recognized and cleaved by the endoribonuclease Dicer.Loading into RISC: the cleaved fragments are incorporated into a multiprotein complex known as the RNA-induced silencing complex (RISC).Guide strand selection: within the RISC, one strand of the siRNA duplex (the guide strand) is selectively retained to direct the complex to its complementary target mRNA.Target recognition: the guide strand base-pairs with the complementary sequence on the target messenger RNA (mRNA).mRNA degradation: RISC, guided by the siRNA, cleaves and degrades the target mRNA, thereby preventing translation [47,48].

siRNAs can be introduced artificially into the cell or can be generated naturally from longer RNA molecules, such as viral RNAs [41].

Following cellular entry and release from the endosomal compartment, siRNA associates with the RNA-induced silencing complex (RISC). Within this complex, one strand of the siRNA duplex, the so-called guide (antisense) strand, is retained, while the passenger (sense) strand is degraded. The guide strand then binds to a complementary messenger RNA (mRNA) that encodes the protein of interest. Upon binding, RISC cleaves and degrades the target mRNA, thereby preventing its translation into protein. This process results in the selective silencing of the target gene [47,48,49,50,51,52] (Figure 2).

Because siRNA is a small, negatively charged molecule, it faces two major challenges:Rapid degradation in the bloodstream due to the activity of nucleases, particularly endonucleases [53,54].Limited cellular uptake, as the negatively charged cell membrane represents a significant barrier to intracellular delivery [25,55].

To address these challenges, several delivery systems have been developed to safely package and transport siRNA into target cells [43,54]. These strategies include:Nanocarriers: nanoscale particles that encapsulate siRNA and facilitate targeted delivery.Aptamers: nucleic acid molecules that specifically recognize and bind to cellular targets.Peptides: short protein fragments that enhance cellular uptake.Sugars and amino-sugars: chemical modifications that improve siRNA stability and cellular recognition.Proteins and antibodies: biomolecules that direct siRNA to specific cell types [43,56,57].

Despite some differences in their modes of action, both antisense oligonucleotides (ASOs) and siRNAs share the same fundamental principle: they bind target RNA molecules through Watson–Crick base pairing and thereby suppress the expression of specific genes [43,58].

## 5. Comparison Between ASO- and siRNA-Based Therapeutics

Although antisense oligonucleotides (ASOs) penetrate cells more efficiently due to their smaller size, double-stranded siRNAs (ds-siRNAs) exhibit greater chemical stability [59]. Moreover, ASOs have been reported to develop pharmacological tolerance, possibly linked to increased pre-mRNA expression [60]. Another important difference is the duration of effect: ASOs generally require more frequent administration compared with siRNAs [58,59,61]. This is explained by the fact that, unlike ASOs, siRNAs activated within RISC can be recycled-the antisense guide strand remains functional and is reused to cleave multiple target mRNA molecules [62].

The main characteristics of ASO- and siRNA-based drugs are summarized in Table 1.

The discovery that double-stranded RNA (dsRNA) is more specific and efficient than single-stranded RNA (ssRNA) in silencing specific RNA sequences subsequently led to the therapeutic exploitation of this mechanism [66,67].

## 6. RNA-Based Therapeutics

Many of the approved RNA-based therapies primarily target rare diseases. Patisiran was the first siRNA-based agent approved by the FDA in 2018 [68], initially indicated for adults with hereditary transthyretin-mediated amyloidosis (hATTR) presenting with polyneuropathy [69,70]. In 2019, givosiran was approved for adults with acute hepatic porphyria, reducing acute attacks and associated symptoms [71,72].

The third siRNA agent, lumasiran, received approval in 2020 for the treatment of primary hyperoxaluria type 1 (PH1) in both children and adults [73]. Lumasiran inhibits the hepatic enzyme glycolate oxidase, thereby reducing urinary oxalate formation and the risk of nephrocalcinosis and kidney stone formation [74,75]. In 2021, inclisiran was approved for adults with heterozygous familial hypercholesterolemia to lower LDL cholesterol [75,76,77]. Inclisiran acts by inhibiting hepatic PCSK9 synthesis and is used as an adjunct in patients who do not reach lipid targets with statins and/or ezetimibe [78,79].

More recently, vutrisiran obtained FDA approval in 2022 for the treatment of polyneuropathy associated with hereditary transthyretin amyloidosis in adults. This subcutaneously administered siRNA therapy offers a simplified dosing regimen compared with patisiran [80,81]. Subsequently, in 2023, nedosiran was approved for primary hyperoxaluria type 1 in children aged ≥9 years and adults [82]. Nedosiran inhibits hepatic lactate dehydrogenase, reducing the conversion of glycolate to oxalate and thereby decreasing renal oxalate burden [83,84]. A summary of these RNA-based therapeutics is provided in Table 2.

Currently, several siRNA molecules are under clinical investigation: fitusiran (for hemophilia A and B) [85,86], teprasiran (for prophylaxis of acute kidney injury post-transplant or after cardiovascular surgery) [87,88], cosdosiran (for anterior ischemic optic neuropathy and glaucoma) [89,90], tivanisiran (for ocular pain and dry eye syndrome) [91], and zilebesiran for arterial hypertension [92].

## 7. Mechanism of Action of Zilebesiran

Zilebesiran is an innovative siRNA, the first in its class, which targets angiotensinogen (AGT) [93]. It represents a promising novel antihypertensive therapy by efficiently silencing the *AGT* gene, thereby reducing angiotensinogen synthesis and subsequently lowering blood pressure and circulating angiotensinogen levels. The structural formula is depicted in Figure 3.

The trivalent GalNAc conjugate enables chemically modified zilebesiran to be preferentially taken up by hepatocytes via the asialoglycoprotein receptor (ASGPR), a receptor on the hepatocyte surface that recognizes glycoproteins lacking sialic acid residues (asialoglycoproteins) [94,95]. Conjugation with trivalent N-acetylgalactosamine (GalNAc) is a sophisticated drug delivery strategy, particularly for liver-targeted therapeutics [96]. This approach exploits the high affinity of GalNAc for ASGPR on hepatocytes, enhancing both delivery efficiency and therapeutic efficacy [97,98]. The trivalent form of GalNAc increases avidity and specificity, which is crucial for effective hepatocyte targeting [25,99,100].

After internalization via clathrin-mediated endocytosis, the siRNA is released into the cytoplasm and incorporated into the RNA-induced silencing complex (RISC) [101]. Within RISC, the guide strand recognizes and degrades the mRNA encoding angiotensinogen, thereby blocking its synthesis and reducing angiotensin II levels, resulting in a direct antihypertensive effect [92,102].

## 8. Early RAAS Inhibition Strategy and Potential Advantages of Zilebesiran

Targeting the renin–angiotensin–aldosterone system (RAAS) at its early stages offers two theoretical advantages compared with classical RAAS inhibitors.

First, since angiotensinogen (AGT) is the precursor of all peptides in the angiotensin pathway, suppressing its expression could lead to broad inhibition of RAAS activity, thereby limiting the “angiotensin II escape” phenomenon that can occur with angiotensin-converting enzyme inhibitors (ACEIs) or angiotensin receptor blockers (ARBs) [103,104]. Even in the presence of compensatory increases in renin secretion, the absence of the substrate (AGT) prevents the formation of angiotensin I and, consequently, angiotensin II [105].

Second, the use of siRNA molecules conjugated with N-acetylgalactosamine (GalNAc) enables liver-specific targeting, thereby minimizing the impact on extrahepatic sources of angiotensinogen, such as renal or adipose tissues [106]. This hepatic specificity may help reduce systemic adverse effects, including renal effects [107]. Preclinical studies have demonstrated near-complete inhibition of hepatic AGT mRNA expression without significantly affecting renal expression [108,109] (Figure 4).

Importantly, unlike ACE inhibitors, zilebesiran does not interfere with bradykinin degradation, suggesting a lower risk of dry cough or angioedema, adverse effects frequently associated with bradykinin accumulation during ACEI therapy [110]. By selectively targeting a central and early component of the RAAS cascade, zilebesiran emerges as a promising agent for sustained blood pressure control.

## 9. Pharmacokinetics and Pharmacodynamics

The pharmacokinetic profile of zilebesiran aligns with its therapeutic objective: prolonged activity primarily localized to the liver. Following subcutaneous administration, peak plasma levels of the small interfering RNA (siRNA) are rapidly achieved, followed by efficient uptake into hepatocytes. Unconjugated siRNA typically has an extremely short circulation half-life of only a few minutes due to rapid degradation by specific nucleases [111]. However, zilebesiran has been chemically modified for enhanced stability, allowing for prolonged systemic durability and enabling subcutaneous administration at extended intervals (twice yearly) [112].

Zilebesiran is administered subcutaneously every 3 to 6 months, owing to its high metabolic stability and sustained intracellular activity. The GalNAc–siRNA conjugate is efficiently taken up by hepatocytes via ASGPR and trafficked into acidic endosomal compartments, where it remains partially sequestered. Chemical modifications protect the molecule from enzymatic degradation, allowing gradual cytosolic release of active siRNA over several weeks. This slow release supports continuous loading of siRNA into the RNA-induced silencing complex (RISC), resulting in prolonged degradation of hepatic AGT mRNA and sustained suppression of angiotensinogen synthesis [113]. Consequently, a single injection maintains reduced Angiotensin II levels for months, with the extended dosing interval primarily reflecting the molecule’s controlled intracellular release and exceptional metabolic persistence rather than slow absorption from the injection site.

The GalNAc moiety enables liver-specific targeting, enhancing hepatic uptake and minimizing systemic exposure [109,113]. In a phase 1 study, Desai et al. showed that subcutaneous zilebesiran produced sustained, dose-dependent reductions in serum angiotensinogen (AGT) and blood pressure lasting up to six months. Pharmacokinetic analyses demonstrated an approximately linear relationship between dose and systemic exposure across the 10–800 mg range. Doses of ≥200 mg produced significant blood pressure reductions by week 8, with maximal mean decreases at 800 mg of −22.5 ± 5.1 mmHg systolic and −10.8 ± 2.7 mmHg diastolic at week 24 [106].

Following single subcutaneous administration, zilebesiran demonstrates dose-proportional pharmacokinetics across the 10–800 mg range. The mean maximum plasma concentration (C_max_) increases nearly linearly with dose, from approximately 150 ng/mL at 10 mg to 2200–2500 ng/mL at 800 mg, indicating first-order kinetics without evidence of saturation. The time to reach maximum concentration (T_max_) occurs between 8 and 16 h post-dose, with an average value of approximately 12 h, suggesting a relatively slow systemic absorption consistent with subcutaneous depot release [106].

After reaching peak levels, plasma concentrations decline gradually over 48 h, reflecting slow elimination and sustained release from both the injection site and hepatic compartments. This pattern is compatible with first-order elimination and an extended plasma half-life estimated at 3–5 days [106].

The area under the concentration–time curve (AUC) also increases proportionally with dose, supporting a linear and dose-dependent pharmacokinetic profile. Although the plasma exposure declines within days, the pharmacodynamic effects—namely, suppression of hepatic angiotensinogen synthesis—persist for several months. This prolonged effect is attributed to the intracellular mechanism of action, where zilebesiran-derived siRNA remains active within the RISC and endosomal compartments, sustaining AGT mRNA silencing long after systemic clearance. All these data are summarized in Table 3.

The evaluation of multiple dosing has shown that zilebesiran pharmacokinetics and pharmacodynamics remain stable under repeated administration. In a phase 1 sub-study, two consecutive 800 mg doses given 12 weeks apart in obese patients produced profiles similar to a single dose [108]. Higher doses resulted in proportional increases in plasma and hepatic concentrations without evidence of excessive or unexpected accumulation, and variations in body mass index did not substantially affect drug exposure or effects [28,107]. These findings indicate that zilebesiran pharmacokinetics remain stable under repeated administration, without significant changes, and that its effects are not substantially influenced by variations in body mass index.

The pharmacodynamic effect of zilebesiran is defined by potent and sustained inhibition of circulating angiotensinogen, leading to corresponding reductions in blood pressure. In the phase 1 study, doses ≥ 100 mg produced >90% reductions in serum angiotensinogen within a few weeks post-administration [106,114]. Mean angiotensinogen reductions exceeded 90% between weeks 3 and 12 following a single dose of ≥100 mg. At the highest tested dose (800 mg), serum AGT suppression remained >90% up to 24 weeks post-dose [106]. According to data reported in the Supplement to Desai AS et al., Effects of zilebesiran on RAAS Markers at Week 12, zilebesiran produced dose-dependent modulation of the renin–angiotensin–aldosterone system (RAAS). At low doses (10–25 mg), inhibition of RAAS activity was minimal, with only minor changes in plasma angiotensin I and II levels and negligible effects on aldosterone or angiotensinogen (AGT). Intermediate doses (50–200 mg) resulted in moderate suppression of AGT synthesis, accompanied by variable reductions in Ang I and Ang II concentrations. In contrast, high doses (400–800 mg) produced marked hepatic AGT suppression (approximately 95–98%), leading to consistent decreases in Ang I and Ang II and modest reductions in aldosterone levels. Notably, Ang II was not completely eliminated, suggesting the persistence of residual enzymatic activity. Plasma renin activity (PRA) showed a slight dose-related increase, consistent with a compensatory negative feedback response [106]. These findings are summarized in Table 4.

The prolonged effect of zilebesiran reflects extended intracellular retention of siRNA in hepatocytes and continuous recycling of AGT mRNA via the RNA-induced silencing complex (RISC) [62]. Active doses (≥200 mg) produced measurable reductions in plasma renin activity, angiotensin I, angiotensin II, and aldosterone, indicating effective inhibition of downstream RAAS components [106].

Regarding elimination, pharmacokinetic studies showed that only a minor fraction of administered zilebesiran (~9–22%) is excreted unchanged in the urine, suggesting renal excretion is a minor pathway. Additionally, the main metabolite, AS(N-1)3′ zilebesiran, was detected at very low concentrations in urine, indicating predominantly intracellular degradation. Thus, zilebesiran is cleared through a combination of intracellular metabolism and minor renal excretion, without significant involvement of hepatic pathways or cytochrome P450 enzymes. This pharmacokinetic profile supports infrequent dosing while maintaining effective systemic levels over extended periods [115].

Blood pressure reductions observed following zilebesiran administration are proportional to the degree of angiotensinogen inhibition. In the phase 1 clinical study [114], higher doses were associated with greater reductions in 24-h ambulatory blood pressure, with a significant negative correlation between dose and change in systolic blood pressure (r = −0.4) [106]. Zilebesiran’s antihypertensive effect became detectable within 2–4 weeks post-dose and persisted up to 12–24 weeks for higher doses. For example, in a cohort of eight subjects receiving 800 mg, mean systolic and diastolic blood pressure reductions at 24 weeks were approximately −22.5 mmHg and −10.8 mmHg, respectively [106]. These sustained responses reflect an extended pharmacodynamic half-life, indicating prolonged clinical effect after a single dose.

## 10. Preclinical Studies

Preclinical studies have further elucidated the reversible mechanisms through which zilebesiran induces blood pressure reduction in rodent models, providing insights into potential clinical management strategies. The blood pressure-lowering effect mediated by AGT-targeted siRNA can be rapidly reversed through administration of angiotensin II or norepinephrine, or gradually reversed via interventions such as fludrocortisone or increased salt intake. This comprehensive understanding of zilebesiran’s mechanism of action and its reversible effects further underscores its potential as a groundbreaking therapy for hypertension [116].

Recently, the REVERSIR (RVR) technology has been developed, utilizing antisense oligonucleotides to block siRNA-mediated mRNA silencing. In animal models, this approach allowed reversible control of zilebesiran’s effects, enabling restoration of angiotensinogen production when needed [37,117]. In spontaneously hypertensive rats, administration of AGT-RVR at doses of 1, 10, or 20 mg/kg reversed the antihypertensive effects of zilebesiran (AGT siRNA, 10 mg/kg), restoring blood pressure to baseline levels within 4–7 days. Higher RVR doses (10 and 20 mg/kg) fully normalized AGT and renin levels, and subsequent administration of a second zilebesiran dose after AGT-RVR again lowered blood pressure proportionally to the prior RVR dosage [37].

### 10.1. Anti-AGT siRNA: Blood Pressure Reduction and Target Organ Protection in Preclinical Models

A preclinical study conducted by Merck evaluated an siRNA encapsulated in lipid nanoparticles, designed to inhibit the *AGT* gene, administered intravenously to male spontaneously hypertensive rats at a dose of 3 mg/kg. The treatment led to a rapid decrease in hepatic AGT mRNA expression and circulating AGT levels, with effects sustained for approximately one week. Systolic blood pressure decreased significantly, reaching a nadir on day 4 post-administration (154 ± 4 mmHg vs. 188 ± 2 mmHg in the control group). Liver and kidney function markers remained within normal limits, with no signs of toxicity [118].

In another preclinical study conducted by Alnylam, a GalNAc-conjugated siRNA targeting the *AGT* gene was tested in spontaneously hypertensive rats (SHR; systolic blood pressure ≥ 150 mmHg) as hypertensive models, and in normotensive rats (systolic blood pressure 110–125 mmHg) as controls, both as monotherapy and in combination with captopril or valsartan. Monotherapies reduced mean arterial pressure by −14 mmHg (siRNA), −23 mmHg (captopril), and −10 mmHg (valsartan), while the combination of valsartan + siRNA had the greatest effect (−68 mmHg). siRNA administration reduced plasma AGT levels by up to 97–99%, and angiotensin I suppression was maintained when combined with valsartan. Furthermore, the valsartan + siRNA combination ameliorated cardiac hypertrophy, evidenced by decreased NT-proBNP levels, while renal function remained stable across all groups [107].

A separate study in hypertensive rats compared daily oral captopril with the RNA-based therapy GalNAc-siRNA R0797070. R0797070 demonstrated a stronger impact on reducing heart mass and cardiac remodeling, with a 10% decrease in heart weight and a 17% reduction in left ventricular myocyte size compared to captopril. These findings suggest that zilebesiran, which shares the same pharmacologic mechanism, may offer superior cardiovascular protection, not only through blood pressure control but also by preventing hypertension-associated cardiac remodeling [119].

Although preclinical studies in rat models of preeclampsia have not shown harmful effects on offspring [120], concerns regarding potential off-target effects remain. Haase et al. investigated hepatic AGT-targeted siRNA therapy for ameliorating preeclampsia symptoms in two pregnant rat models. The first model used transgenic rats expressing human AGT and renin, leading to uteroplacental renin–angiotensin–aldosterone system (RAAS) activation. The second model, RUPP (Reduced Uterine Perfusion Pressure), induced ischemia/reperfusion and local/systemic inflammation. In both models, siRNA treatment significantly reduced blood pressure and proteinuria and improved fetal growth, without adverse effects on the fetus or placenta [121].

Therefore, further studies are needed to assess whether zilebesiran can cross the placenta and exert similar effects on fetal development. Importantly, zilebesiran’s hepatocyte-targeted delivery system preserves extrahepatic AGT expression, limiting potential effects on the kidneys and adipose tissue, which are relevant AGT sources, particularly in obese individuals [122,123,124].

### 10.2. Anti-AGT siRNA Therapy in Ocular Diseases: Preclinical Evidence

Consistent evidence demonstrates the presence of renin-angiotensin system (RAS) components in numerous ocular structures, where they can influence the development of eye diseases. Dysregulation of systemic or local RAS may contribute to ocular pathology, and pharmacologic modulation of this system provides opportunities for developing novel or adjunctive therapies in eye disorders.

In patients with primary open-angle glaucoma (POAG), levels of angiotensin-converting enzyme (ACE) in tears are elevated compared to non-glaucomatous controls [125,126,127]. Angiotensin II exerts effects on retinal circulation similar to those observed in other organs. Additionally, Ang II stimulates pericyte migration and proliferation and induces VEGF mRNA expression, processes that can promote neovascularization. The ACE inhibitor captopril and the angiotensin receptor blocker (ARB) candesartan have demonstrated the ability to normalize impaired retinal blood flow in rodent models of diabetic retinopathy [128,129,130].

Multiple studies have shown that RAS inhibitors reduce the risk of developing diabetic retinopathy and increase the likelihood of disease regression [126,131,132]. A study in diabetic retinopathy rat models demonstrated that inhibition of angiotensinogen (AGT) using siRNA-AGT reduced activation of the Ang II–ERK1/2 signaling pathway, promoted proliferation, and decreased apoptosis of retinal endothelial cells. These effects were similar to those achieved using miR-133b mimetics and contrasted with the worsening observed after miR-133b inhibition.

These findings suggest that siRNA-AGT may represent a promising therapeutic strategy to limit the progression of diabetic retinopathy through local modulation of the renin-angiotensin system [127].

### 10.3. RAS Modulation in Hepatic Fibrosis and Portal Hypertension

The renin-angiotensin system (RAS) plays a complex and significant role in liver disease pathophysiology, influencing both hepatic fibrosis and portal hypertension. Its effects extend beyond traditional systemic functions [133]. Recently, the discovery of an alternative local RAS pathway in the liver has opened new perspectives for understanding mechanisms involved in chronic liver disease progression. Inhibition of RAS through suppression of hepatic angiotensinogen production may represent a promising therapeutic approach, with potential to reduce hepatic fibrosis and portal hypertension [134]. This strategy highlights a novel therapeutic avenue in the management of liver diseases, offering innovative opportunities that warrant further exploration.

### 10.4. Renoprotective Effects of Anti-AGT siRNA Therapy in Diabetic Models

Small interfering RNA (siRNA) therapy targeting hepatic angiotensinogen (AGT) has demonstrated significant renal protective effects in animal models of diabetes. Studies in TGR (mRen2) diabetic rats showed that siRNA administration, either alone or in combination with valsartan, markedly reduced albuminuria and proteinuria, while improving glomerulosclerosis and podocyte dysfunction, indicating a protective effect on renal structure and function [135].

Although all tested treatments reduced mean arterial pressure (MAP) and cardiac hypertrophy, only therapies including siRNA, with or without valsartan, as well as the combination of valsartan and captopril, effectively reduced renal angiotensin I and II levels. This underscores the importance of suppressing liver-derived AGT in renal protection, an effect that appears to be independent of blood pressure changes [136]. These findings suggest that targeting hepatic AGT production may represent an innovative and effective strategy to mitigate diabetes-associated renal injury, regardless of its impact on blood pressure control.

## 11. Clinical Studies

### 11.1. Phase 1 Clinical Trial

A key pharmacodynamic feature of zilebesiran is that its effect is influenced by dietary sodium intake and concomitant RAAS inhibitor therapy. In patients with high sodium intake, the effect of zilebesiran is reduced, likely due to enhanced activation of the renin–angiotensin–aldosterone system, which counteracts the angiotensin-lowering action of zilebesiran. Conversely, under conditions of sodium restriction or RAAS inhibition, its antihypertensive effect may be more pronounced [113]. In a controlled salt intake sub-study (Phase 1, Part B) [106], a high-sodium diet significantly reduced the antihypertensive effect of zilebesiran, whereas a low-sodium diet enhanced it [137,138]. Conversely, co-administration with an angiotensin receptor blocker (ARB), irbesartan (Phase 1, Part E) [106], produced greater blood pressure reductions than zilebesiran alone, indicating additive or synergistic effects by combining upstream (AGT reduction) and downstream (receptor blockade) RAAS inhibition. These pharmacodynamic data are essential for guiding zilebesiran use, emphasizing the need for dietary sodium restriction for maximal efficacy and confirming the safety of combining zilebesiran with other RAAS inhibitors for enhanced therapeutic effect [105].

In a phase 1 clinical study conducted by Desai et al., 107 patients with hypertension were randomized in a 2:1 ratio to receive either a single subcutaneous dose of zilebesiran (10–800 mg, dose-escalation) or placebo, with follow-up over 24 weeks (Part A). Part B evaluated the effect of an 800 mg dose on blood pressure under low- or high-sodium dietary conditions, while Part E investigated the impact of concomitant administration of the same zilebesiran dose with irbesartan.

Zilebesiran’s antihypertensive effect became detectable within 2–4 weeks post-administration and persisted up to 12–24 weeks for higher doses. After a single dose of ≥200 mg, mean systolic blood pressure decreased by >10 mmHg and diastolic pressure by >5 mmHg at week 8. For instance, in a cohort of eight subjects receiving 800 mg, mean reductions at week 24 were approximately −22.5 mmHg for systolic and −10.8 mmHg for diastolic blood pressure [108]. At doses ≥ 200 mg, significant blood pressure reductions were observed as early as week 8, with maximal mean reductions at the 800 mg dose of −22.5 ± 5.1 mmHg (systolic) and −10.8 ± 2.7 mmHg (diastolic) at week 24 [106]. The study design, including Parts A, B, and E, is illustrated in Figure 5.

Results from Parts B and E demonstrated attenuation of the antihypertensive effect with a high-sodium diet and potentiation with co-administration of irbesartan. In Part B, after one week of a low-sodium diet prior to zilebesiran administration, systolic and diastolic blood pressure decreased by −9.1 mmHg and −2.4 mmHg, respectively, returning to baseline under a high-sodium diet. Administration of 800 mg zilebesiran amplified these reductions, with systolic and diastolic decreases of −18.8 mmHg and −8.4 mmHg, respectively, again returning to baseline under a high-sodium diet [106].

In Part E, at week 6, patients receiving 800 mg zilebesiran alone exhibited a mean systolic blood pressure reduction of −21.8 ± 2.9 mmHg. Patients with persistent ambulatory systolic BP ≥ 120 mmHg who received additional irbesartan experienced further reductions of −6.3 ± 3.1 mmHg (systolic) and −3.0 ± 1.9 mmHg (diastolic). These findings indicate that zilebesiran is effective as monotherapy and can be safely combined with other antihypertensives for optimal blood pressure control. Blood pressure reductions were sustained during both daytime and nighttime periods, while maintaining the physiological circadian profile. The nighttime “dipping” pattern was preserved or slightly enhanced, without evidence of excessive nocturnal hypotension or flattening of the circadian curve. Throughout the 24-week follow-up, hepatic and renal function were closely monitored, with no clinically significant changes observed in serum creatinine or estimated glomerular filtration rate. Although unmodified siRNAs may activate innate immune sensors such as TLR3, modern GalNAc-conjugated and chemically stabilized siRNA molecules exhibit minimal immunogenicity. In phase-1 studies of zilebesiran, anti-drug antibodies were detected in approximately 2% of participants, were low-titer and transient, and had no apparent clinical relevance. These findings support that immunogenic risk is effectively mitigated through rational chemical design and GalNAc-mediated hepatocyte targeting [28,106].

### 11.2. Phase II Clinical Studies

Efficacy and Safety of zilebesiran as Monotherapy in Mild-to-Moderate Hypertension

The phase II KARDIA-1 study included 394 patients with mild to moderate hypertension, randomized to receive subcutaneous zilebesiran (150, 300, or 600 mg every 6 months, or 300 mg every 3 months) or placebo [139]. After 3 months of treatment, all four zilebesiran dosing regimens produced significant reductions in mean 24-h ambulatory systolic blood pressure (the primary endpoint), with decreases ranging from 7.3 to 10 mmHg from baseline, compared with a mean increase of 6.8 mmHg in the placebo group [28,139].

Specifically, at 3 months, mean reductions in 24-h ambulatory systolic BP were −7.3 mmHg (150 mg/6 months), −10.0 mmHg (300 mg/3–6 months), and −8.9 mmHg (600 mg/6 months), versus a +6.8 mmHg increase in the placebo group. Adjusted differences from placebo were −14.1 mmHg, −16.7 mmHg, and −15.7 mmHg, respectively (*p* < 0.001). At 6 months, reductions in ambulatory systolic BP were sustained across all zilebesiran groups, while the placebo group exhibited increases, with statistically significant differences maintained for all doses and regimens (*p* < 0.001). Office and ambulatory systolic BP reductions ranged from −9.1 to −12.0 mmHg versus placebo at both 3 and 6 months (*p* < 0.001), with higher responder rates observed in all treated groups [139,140].

Zilebesiran’s antihypertensive effect was durable across the tested dosing intervals of 3 to 6 months. The 300 mg and 600 mg regimens achieved sustained reductions in ambulatory systolic blood pressure and maintained angiotensinogen (AGT) suppression of >90%, comparable to the 800 mg dose in the prior phase I study [113]. These findings indicate that lower doses may be both effective and pharmacodynamically potent [108,120,122,123]. In the KARDIA-1 study, 24-h ambulatory systolic blood pressure (SBP) was assessed. All treatment regimens produced significant and sustained reductions in SBP compared with placebo, while preserving the physiological circadian pattern of blood pressure. The nighttime “dipping” profile was maintained or slightly enhanced in the treated groups, indicating effective blood pressure control over the full 24-h period without disruption of the day–night rhythm. Reductions were consistent during both daytime and nighttime, supporting the maintenance of a normal circadian blood pressure profile. These findings reinforce observations from phase 1 studies, which demonstrated preservation of the nighttime dipping pattern, confirming that the intervention not only lowers blood pressure but also maintains the circadian architecture of blood pressure regulation. Blood pressure reductions were consistent throughout the 24-h period, and diastolic BP decreases were proportional across all treatment arms. The results demonstrate that intermittently administered, subcutaneous therapy can provide clinically meaningful, durable reductions in blood pressure [139].

### 11.3. Efficacy and Safety of Zilebesiran as Add-On Therapy in Uncontrolled Hypertension

Given that most patients with hypertension require the combination of two or more medications to achieve therapeutic targets, it is essential to evaluate the efficacy and safety of zilebesiran in combination with other established classes of antihypertensive agents [141]. This was addressed in the KARDIA-2 study, a phase II clinical trial designed to assess the efficacy and safety profile of zilebesiran administered as adjunctive therapy in patients with inadequately controlled hypertension despite standard antihypertensive treatment.

The study included adults with mild to moderate hypertension who were either previously untreated or on stable antihypertensive therapy with a maximum of two medications, with a mean age of 58.5 years. After discontinuation of prior therapy, patients were randomized in a 10:7:4 ratio to receive one of the following once-daily oral antihypertensive regimens: olmesartan 40 mg, amlodipine 5 mg, or indapamide 2.5 mg. Following at least four weeks of treatment according to protocol, patients with a 24-h mean ambulatory systolic blood pressure (SBP) between 130 and 160 mmHg were further randomized in a double-blind, 1:1 ratio to receive a single subcutaneous dose of zilebesiran 600 mg or placebo as adjunctive therapy [142,143].

Administration of a single dose of zilebesiran was associated with a significant reduction in 24-h mean ambulatory systolic blood pressure at three months, compared with placebo. Adjusted mean differences from placebo were −4.0 mmHg in the olmesartan-treated group, −9.7 mmHg in the amlodipine-treated group, and −12.1 mmHg in the indapamide-treated group, all differences being statistically significant (*p* < 0.05) [102,142,143]. The therapeutic response was generally consistent across most analyzed subgroups; however, a trend toward a slightly reduced antihypertensive effect was observed in Black patients [142,143]. These findings are consistent with known physiological and genetic differences in the RAAS among African American individuals. From childhood, they tend to exhibit higher sodium retention, leading to chronically suppressed renin and lower angiotensin I and II levels, a ‘low-renin’ state that limits sodium loss [121,144]. Additionally, genetic variants affecting renal sodium reabsorption and aldosterone secretion (e.g., SCNN1B, NEDD4, ARMC5) may represent evolutionary adaptations to low-sodium, high-temperature environments [145,146]. These factors likely contribute to the reduced responsiveness of African American patients to RAAS-targeted therapies such as zilebesiran.

Zilebesiran demonstrated a favorable profile, with a low incidence of adverse events across all treatment arms. Administration of zilebesiran resulted in a significant 24-h reduction in systolic blood pressure of −15.12 mmHg and diastolic blood pressure of −7.34 mmHg at 12 weeks compared with placebo, with consistent results across all studies (I^2^ = 0%) [142,143,147].

### 11.4. Efficacy and Safety of Zilebesiran as Add-On Therapy in Uncontrolled Hypertension with High Cardiovascular Risk

More recently, once the efficacy and safety of zilebesiran in combination with other established antihypertensive agents had been demonstrated, its investigation was extended to patients with various comorbidities, including chronic kidney disease (CKD), a population at high risk for difficult-to-control hypertension [148].

In this context, the KARDIA-3 study is a randomized, double-blind, placebo-controlled trial that included two cohorts: Cohort A, with an estimated glomerular filtration rate (eGFR) of at least 45 mL/min/1.73 m^2^, and Cohort B, with an eGFR between 30 and <45 mL/min/1.73 m^2^ [149,150]. Participants were randomized to receive a single subcutaneous dose of zilebesiran (150, 300, or 600 mg) or placebo. A total of 270 patients were included in the analysis, with a median age of 67 years, of whom 45% were female; 23% had known cardiovascular disease, and 77% were considered at high cardiovascular risk. Baseline blood pressure values were, on average, 144/80 mmHg in-office and 142/79 mmHg on 24-h ambulatory monitoring. In Cohort A, three months after administration of a single subcutaneous dose of zilebesiran, the placebo-adjusted reduction in systolic blood pressure (SBP) was −5.0 mmHg (95% CI: −9.9; −0.2) for the 300 mg dose and −3.3 mmHg (95% CI: −8.2; 1.6) for the 600 mg dose, differences that did not reach statistical significance after multiple testing. At six months, the placebo-adjusted in-office SBP reduction was −3.9 mmHg for zilebesiran 300 mg and −3.6 mmHg for 600 mg, whereas the reduction in ambulatory blood pressure was more pronounced, at −5.5 mmHg and −7.4 mmHg, respectively. The effect on nocturnal blood pressure was even greater, with reductions of −6.6 mmHg (300 mg) and −8.2 mmHg (600 mg) [149,150]. Results from the KARDIA-3 Cohort B are expected to be presented at an upcoming medical meeting.

Alnylam, in collaboration with Roche, plans to initiate by the end of 2025 the global Phase III ZENITH CVOT study, designed to evaluate the efficacy and safety of zilebesiran in the prevention of major cardiovascular events, with study completion expected in 2030. The study will enroll approximately 11,000 patients from over 30 countries with uncontrolled hypertension despite treatment with at least two standard antihypertensive agents (including one mandatory diuretic), and with established cardiovascular disease or high cardiovascular risk [28].

The primary objective of the trial will be to assess the impact of zilebesiran 300 mg on the reduction in the risk of cardiovascular death, non-fatal myocardial infarction, non-fatal stroke, and heart failure events (hospitalization or urgent visit), compared with placebo. Unlike some conventional medications, which may fail to maintain efficacy over a full 24-h period due to their pharmacokinetic and pharmacodynamic properties or suboptimal dosing [151], zilebesiran emerges as a potentially revolutionary therapy [102].

## 12. Adverse Effects

Although the therapeutic effects of zilebesiran are promising, they underscore the need to carefully evaluate its safety profile, particularly with respect to subcutaneous administration, which can be associated with local or systemic adverse reactions. In the Phase I study conducted by Desai et al., the most common adverse events were injection-site reactions (erythema or pigmentation), along with headache and upper respiratory tract infections [106]. Anti-drug antibodies, generally low-titer and transient, were detected in 2 of 80 patients (2%) in the same Phase I study [106].

In the KARDIA-1 study, the most frequent treatment-related adverse event was injection-site reaction, reported in 6.3% (*n* = 19) of patients. All were mild or moderate and transient, with pain and erythema as predominant symptoms. Patients receiving zilebesiran every 3 months experienced more injection-site reactions than those treated every 6 months [139]. Other drug-related adverse events included hyperkalemia 5.3% (*n* = 16), hypotension 4.3% (*n* = 13), acute kidney injury 1.3% (*n* = 4), and transient hepatic events 3.0% (*n* = 9), most of which were mild and resolved spontaneously without treatment discontinuation. Only four patients permanently discontinued the drug due to orthostatic hypotension [28].

In the KARDIA-2 study, adverse events such as injection-site reactions were reported in 3% (*n* = 10) of patients. Hyperkalemia 5.5% (*n* = 18), hypotension 4.3% (*n* = 14), and acute kidney injury 4.9% (*n* = 16) occurred at rates similar to previous studies, with most being mild and self-limited. Zilebesiran caused >30% reductions in estimated glomerular filtration rate (eGFR) within the first 3 months; however, these changes resolved spontaneously without medical intervention [142,143].

In the KARDIA-3 study, most adverse events were mild or moderate, non-serious, and transient, with only a few cases requiring intervention. The incidence of hyperkalemia, renal impairment, and hypotension was low [149].

Comparative analyses with other siRNA therapies, such as lumasiran and givosiran, reveal distinct adverse event patterns. Injection-site reactions are particularly common with lumasiran, whereas givosiran is associated with fatigue, nausea, and chronic kidney disease [152,153]. Patisiran, another siRNA therapy, frequently causes mild-to-moderate infusion-related reactions and is used in cardiology for cardiac amyloidosis [154].

Overall, zilebesiran presents a unique safety profile, with injection-site reactions representing the primary adverse event. Importantly, it has demonstrated low rates of adverse events, with the most common being mild injection-site reactions and no clinically relevant changes in renal or hepatic function, as highlighted by Bakris et al. [139] (Table 5).

## 13. Future Directions

Zilebesiran represents a paradigm shift in the management of hypertension. Its subcutaneous administration at 3–6 months intervals provides sustained blood pressure reductions, offering a promising option for patients with adherence challenges or polypharmacy. This innovative approach may be particularly useful in resistant or uncontrolled hypertension, both as monotherapy and as add-on therapy. Preclinical and clinical studies have demonstrated that zilebesiran provides therapeutic benefits, including blood pressure reduction and organ protection.

Beyond its antihypertensive effect, zilebesiran may contribute to the prevention of systemic complications of hypertension, including cardiovascular, renal, or retinal damage. However, further studies are needed to evaluate long-term safety, considering potential adverse effects identified in preclinical studies, such as hematopoietic effects, and to determine safety in pregnant or breastfeeding women. Additionally, zilebesiran could be beneficial in high cardiovascular risk populations, not only by lowering blood pressure but also by preventing target organ damage. The KARDIA-3 study, which includes patients with uncontrolled hypertension and elevated cardiovascular risk, will provide further data on the clinical utility of zilebesiran in this challenging patient population.

Multiplicity-adjusted analyses at 3 months did not reach statistical significance. Potential explanations for the attenuation of treatment effects in high-risk, multi-drug cohorts include ceiling effects, where patients already receiving intensive antihypertensive therapy have limited capacity for further blood pressure reduction, and the influence of concomitant medications, such as diuretics or ACE inhibitors, which may alter RAAS activity and reduce the incremental efficacy of zilebesiran.

Aprocitentan, a dual endothelin receptor antagonist (ETA/ETB), exerts its effects by inhibiting the vasoconstrictive and proinflammatory actions mediated by endothelin-1 (ET-1), thereby contributing to blood pressure reduction and attenuation of adverse vascular remodeling. Experimental and clinical studies have demonstrated a dose-dependent antihypertensive effect, a prolonged duration of action (half-life of approximately 44 h), and good tolerability when combined with renin–angiotensin system (RAS) blockers such as enalapril or valsartan [155,156]. In animal models of hypertension, including sodium-depleted conditions, the combination of aprocitentan with RAS inhibitors produced an additional reduction in blood pressure without renal impairment and with a superior safety profile compared with spironolactone, avoiding the increased risk of hyperkalemia and renal dysfunction associated with excessive RAS blockade [157,158]. Given the complementary mechanisms of action of aprocitentan (endothelin system blockade) and zilebesiran (hepatic inhibition of angiotensinogen synthesis leading to upstream suppression of the entire RAS cascade), the concomitant use of these two agents could provide enhanced blood pressure control while maintaining a favorable safety profile. Future studies evaluating this combination are warranted, as they may define a novel therapeutic strategy based on the synergistic inhibition of two major pathophysiological pathways involved in hypertension.

In the context of the 2024 ESC guidelines (defining hypertension as ≥140/90 mmHg, with lower systolic blood pressure targets when tolerated) and the 2025 AHA/ACC recommendations, zilebesiran may represent a therapeutic option for patients with difficult-to-control hypertension, particularly in cases of polypharmacy or suboptimal adherence to daily oral therapy, owing to its twice-yearly subcutaneous administration. Moreover, zilebesiran could serve as an adjunct to existing antihypertensive regimens (such as ACE inhibitors, ARBs, or diuretics) in patients with incomplete blood pressure control, providing an additional pathway to achieve guideline-recommended targets.

Interindividual variability in response to zilebesiran may partially result from genetic factors. Single-nucleotide variants (SNVs) or small insertions/deletions (indels) within the *AGT* gene or other components of the renin–angiotensin system (e.g., ACE, *AGTR1*, *AGTR2*) could influence baseline angiotensinogen expression or siRNA binding efficiency.

Moreover, polymorphisms affecting the asialoglycoprotein receptor (ASGPR) or hepatic uptake pathways might modulate drug internalization and exposure.

Such genetic variability could contribute to the observed interindividual differences in both efficacy (extent and duration of AGT suppression) and adverse drug reactions.

Future studies incorporating pharmacogenomic profiling may help clarify these associations.

## 14. Conclusions

Zilebesiran represents an innovative antihypertensive therapy that targets the RAAS cascade by inhibiting hepatic angiotensinogen via RNAi technology. Clinical studies KARDIA-1, KARDIA-2, and KARDIA-3 have demonstrated that subcutaneous administration of zilebesiran can produce sustained reductions in blood pressure and angiotensinogen levels, with a favorable safety profile and minimal adverse effects. This approach may offer a significant advantage for patients with resistant or uncontrolled hypertension and for those at high cardiovascular risk, particularly where adherence to daily oral therapies is challenging.

The use of zilebesiran during pregnancy and lactation is not recommended and is considered contraindicated. Preclinical studies in rats using angiotensinogen siRNA in preeclampsia models did not demonstrate adverse effects on offspring, nor was placental transfer detected; however, these findings are insufficient to establish safety in humans.

Beyond blood pressure reduction, zilebesiran has the potential to prevent target organ damage, including the heart, kidneys, and retina. However, further studies are needed to evaluate long-term safety, including in special populations such as pregnant or breastfeeding women. In conclusion, zilebesiran holds promise as a foundational option in the management of difficult-to-control hypertension, with future studies expected to clarify its full clinical and therapeutic impact.

## Figures and Tables

**Figure 1 ijms-26-10717-f001:**
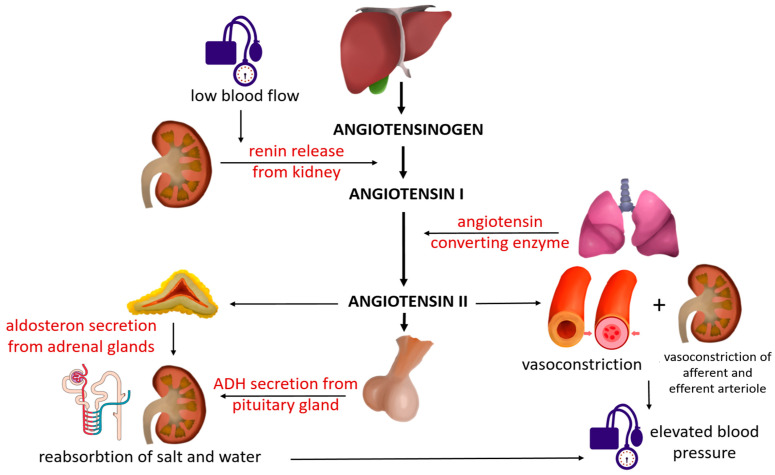
Mechanism of blood pressure regulation through the renin–angiotensin–aldosterone system (*figure created by R. Tudosa*).

**Figure 2 ijms-26-10717-f002:**
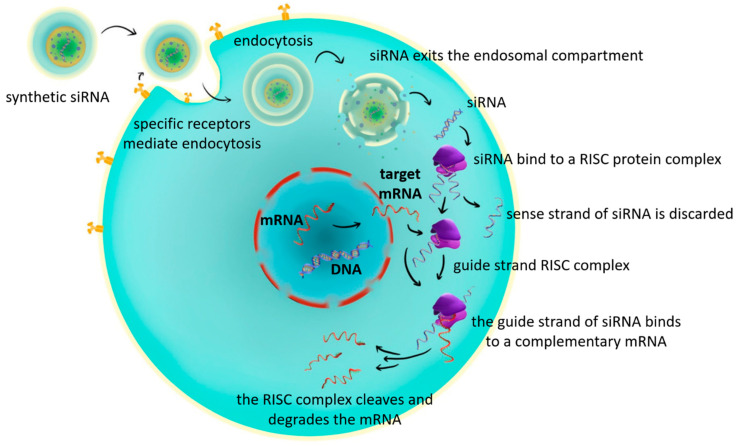
The mechanism of small interfering RNA therapy (*figure created by R. Tudosa*).

**Figure 3 ijms-26-10717-f003:**
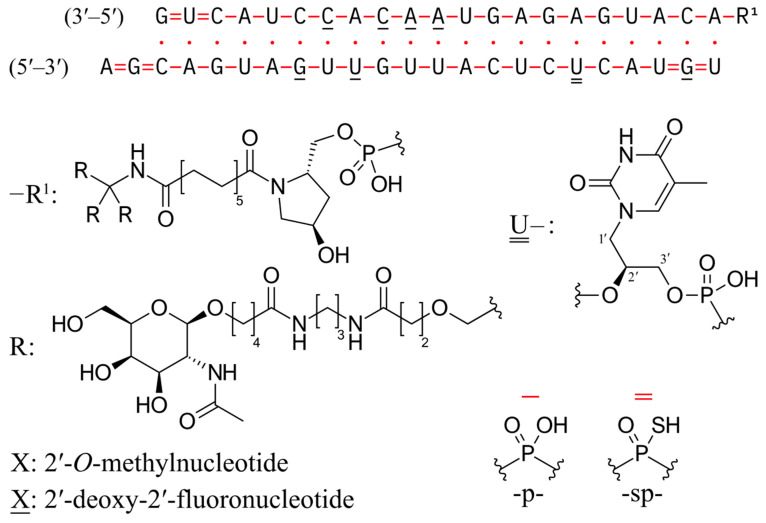
Zilebesiran structural formula (based on WHO Drug Information, 35:3, 2021, public domain).

**Figure 4 ijms-26-10717-f004:**
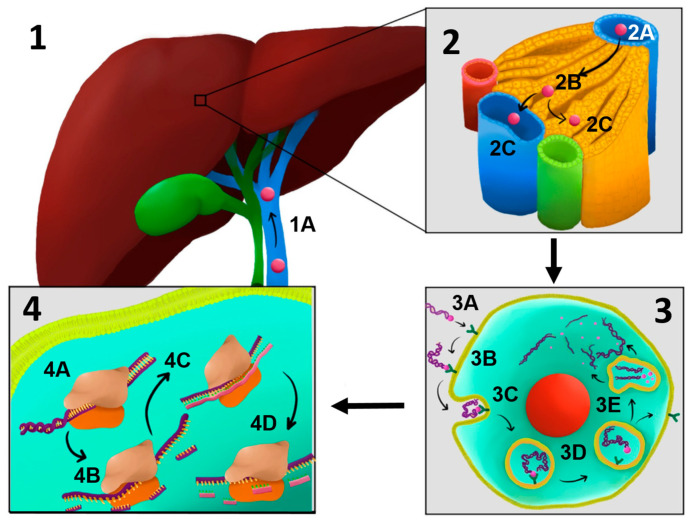
Hepatic Mechanism of Action of Zilebesiran: (**1**) Once administered subcutaneously, zilebesiran reaches the liver via the portal vein (1A). (**2**) Upon reaching the liver (2A), zilebesiran is delivered to hepatocytes (2B), with a portion remaining stored in the liver and another portion entering the systemic circulation via the inferior vena cava (2C). (**3**) Zilebesiran care stored in the liver reaches the hepatocyte (3A). Trivalent GalNAc conjugate enables zilebesiran to selectively bind the ASGR receptor on hepatocytes (3B). Binding to the ASGR receptor triggers clathrin-mediated endocytosis (3C). Zilebesiran is released into the cytoplasm and associates with RISC (3D). (**4**) Zilebesiran binds to the RISC complex (4A). The passenger strand is discarded, and the guide strand directs sequence-specific mRNA cleavage (4B). The guide strand binds to a complementary mRNA (4C). The degradation of mRNA prevents the synthesis of angiotensinogen (4D) (*figure created by R. Tudosa*).

**Figure 5 ijms-26-10717-f005:**
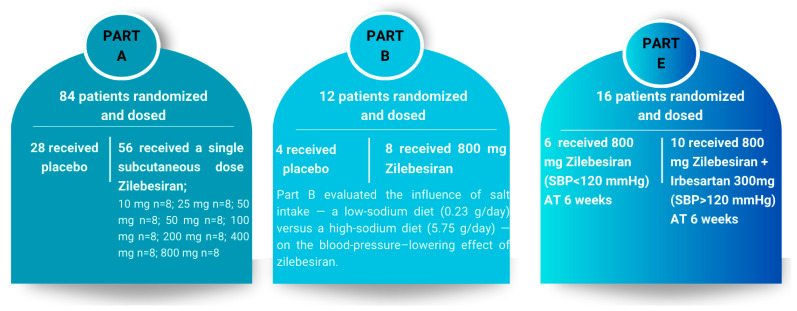
Phase 1 study design based on data reported by Desai et al. [111].

**Table 1 ijms-26-10717-t001:** Key characteristics and differences between antisense oligonucleotides and small interfering RNA.

Characteristic	ASO (Antisense Oligonucleotide)	siRNA (Small Interfering RNA)
Structure	Single-stranded RNA (ssRNA), 8–50 nucleotides	Double-stranded RNA (dsRNA), 20–24 base pairs
Molecular weight (kDa)	~12 [63]	~21 [63]
Mechanism of action	Binds target mRNA → activates RNase H → mRNA degradation [63,64]	Incorporated into RISC → recognizes target mRNA → cleavage [43]
Target location	mRNA (nucleus or cytoplasm)	mRNA (cytoplasm)
Enzyme involved	RNase H [64]	RISC with endonuclease activity [61]
Cellular entry	Endocytosis	Endocytosis with endosomal release
Active strand	Single strand (complementary to mRNA) [65]	Only the antisense (guide) strand is active in RISC
Versatility	Can modify splicing, block translation, or degrade mRNA [58]	Specific for mRNA degradation
Immunogenicity	High	High [61]
Half-life	Months [63]	>1 year [58,64]
Route of administration	Subcutaneous	Subcutaneous
Dosing frequency	Monthly	A few times per year [61]

**Table 2 ijms-26-10717-t002:** Major FDA-approved RNA-based therapeutics.

Drug	Year of FDA Approval	Indication
Patisiran	2018	Polyneuropathy associated with hereditary transthyretin-mediated amyloidosis (hATTR-PN)
Givosiran	2019	Acute hepatic porphyria (AHP)
Lumasiran	2020	Primary hyperoxaluria type 1 (PH1)
Inclisiran	2021	Primary hyperlipidemia, including heterozygous familial hypercholesterolemia (HeFH)
Vutrisiran	2022	Polyneuropathy associated with hereditary transthyretin-mediated amyloidosis
Nedosiran	2023	Primary hyperoxaluria type 1

**Table 3 ijms-26-10717-t003:** Summary of zilebesiran pharmacokinetic and pharmacodynamic parameters, based on data reported by Desai et al. [111].

Parameter	Estimated Value/Observation	Conclusion
C_max_	150–2500 ng/mL (10–800 mg)	Increases proportionally with dose
T_max_	8–16 h	Slow subcutaneous absorption
AUC	Increases proportionally with dose	Linear relationship
Plasma t½	~3–5 days (estimated)	Slow elimination
Kinetic type	First-order	Exponential elimination
Dose dependence	Yes, linear	No saturation observed
Pharmacodynamic ½	3–6 months	Due to the RISC-mediated mechanism

**Table 4 ijms-26-10717-t004:** Effects of zilebesiran on RAAS markers at week 12 based on data reported by Desai et al. [111].

Marker (Units)	Dose Range (mg)	Change vs. Baseline (Week 12)	Interpretation/Trend
Aldosterone (nmol/L)	10–25	+0.019–+0.101	Minor or no increase
	50–200	+0.142 → −0.100	Variable changes
	400–800	−0.071 → −0.094	Moderate decrease/stabilization
Angiotensin I (pmol/L)	10–25	+0.381–+1.175	Slight increase at low doses
	50–200	+1.157–−4.425	Mixed/variable effect
	400–800	−3.481 → −3.342	Clear decrease proportional to AGT suppression
Angiotensin II (pmol/L)	10–25	+0.281–+0.688	Stable/marginal change
	50–200	−0.464 → −0.563	Slight decrease
	400–800	−3.075	Pronounced decrease, not complete ablation
Plasma Renin Activity (ng/mL/h)	All doses	Baseline: 0.079–0.428 → modestly increased at W12	Compensatory feedback activation (↑PRA)

**Table 5 ijms-26-10717-t005:** Summary of clinical trials evaluating subcutaneous zilebesiran in patients with hypertension.

Study	Patients	Population	Intervention	Comparator	Follow-Up Duration	Primary Objectives	Key Results
KARDIA-1	394	Mild-to-moderate hypertension	Subcutaneous zilebesiran, single doses 10–800 mg (Part A), 800 mg with low-/high-salt diet (Part B), 800 mg + irbesartan (Part E)	Placebo	12–24 weeks	Safety, pharmacokinetics/pharmacodynamics, and change in 24-h ambulatory BP	Mean systolic BP reductions: 7.3–10 mmHg; effect detectable at 2–4 weeks, persisted up to 24 weeks; negative dose–response correlation (r = −0.4) [139]
KARDIA-2	672	Mild-to-moderate hypertension, untreated or on 1–2 antihypertensive agents	Subcutaneous zilebesiran 600 mg + existing antihypertensive therapy (olmesartan, amlodipine, indapamide)	Placebo + same background therapy	3–6 months	Efficacy and safety as adjunct therapy	Mean 24-h ambulatory systolic BP reductions: −4.0 mmHg (olmesartan), −9.7 mmHg (amlodipine), −12.1 mmHg (indapamide); sustained effect up to 6 months in amlodipine and indapamide groups [25,142]
KARDIA-3	270	Uncontrolled hypertension + CVD or high risk, Chronic kidney disease	Subcutaneous zilebesiran 150, 300, or 600 mg, single dose	Placebo	6 months	Change in office and ambulatory BP, safety	At 3 months, office SBP: −5.0 mmHg (300 mg), −3.3 mmHg [150]

## Data Availability

No new data were created or analyzed in this study. Data sharing is not applicable to this article.

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
