# Peer review of "Zilebesiran as an Innovative siRNA-Based Therapeutic Approach for Hypertension: Emerging Perspectives in Cardiovascular Medicine"

_ijms, 2025, doi:10.3390/ijms262110717_

Round 1
Reviewer 1 Report
Comments and Suggestions for Authors
General Comments: This is an interesting review of the current status of siRNA technology to prevent the formation of angiotensinogen (AGT) specifically by the liver. Presumably, AGT produced by the kidneys, adipose tissue, and other sites is not affected. The results of clinical trials are discussed, and appropriate conclusions are reached. Overall, the review is well organized and comprehensive, but somewhat tedious. Several sections get into details that stray substantially from the main theme of document.
Specific Comments:
Section 3, line 126, I recommend using "inappropriate" instead of "excessive" activation. Fig 1 is useful but should include direct actions of Ang II on kidney vasculature and tubules. The RAS pathway is Section 3 and 3 is repeated again in the next section on Innovative Therapeutics which should be 4. The figure in this section is helpful, but notice that the line on right states that "siRNA exist the..." Did you mean to say "exits"? The section on "RNA-based Therapeutics" strays away from the main theme by discussion of many other siRNA agents that have been developed. I recommend either deleting or condensing this section to just a brief comment regarding the growing family of these agents and perhaps putting the paragraph at the beginning of the next section. This and subsequent sections provide the essence of the article. However, figures 4 and 5 are not appropriate for a professional journal and could be deleted or replaced by more meaningful figures including some graphs that plot dose and time dependent effects on blood pressure and maybe other parameters measured to complement what is in the text demonstrating how long the effects are sustained between treatments. The text on lines 334-340 is almost an exact repetition of the text in the preceding paragraph covering lines 325-332. In section 10 discussing the Phase 1 clinical trial, it is not clear what is being described when referring to parts A , B, and E. Seems like there is a graph or figure missing that shows these parts. Finally, were there any 24-hour plots showing the blood pressure variations and whether the nighttime dipping pattern was improved or maintained?
Author Response
- Section 3, line 126, I recommend using "inappropriate" instead of "excessive" activation.
Thank you for your suggestion. The term has been revised accordingly, and “excessive” has been replaced with the more appropriate term “inappropriate” activation in Section 3, line 126.
- Fig 1 is useful but should include direct actions of Ang II on kidney vasculature and tubules.
We thank the Reviewer for this constructive comment. As suggested, Figure 1 has been revised to include the direct actions of Angiotensin II on both the renal vasculature and tubular segments. We believe these additions enhance the completeness and clarity of the figure.
- The RAS pathway is Section 3 and 3 is repeated again in the next section on Innovative Therapeutics which should be 4. The figure in this section is helpful, but notice that the line on right states that "siRNA exist the..." Did you mean to say "exits"?
We thank the reviewer for pointing this out. The section numbering has been corrected, and the typographical error in the figure legend (“exist” → “exits”) has been fixed.
- The section on "RNA-based Therapeutics" strays away from the main theme by discussion of many other siRNA agents that have been developed. I recommend either deleting or condensing this section to just a brief comment regarding the growing family of these agents and perhaps putting the paragraph at the beginning of the next section. This and subsequent sections provide the essence of the article.
We thank the reviewer for this insightful comment. One of the reviewers specifically requested further elaboration of the “RNA-based Therapeutics” section.
- However, figures 4 and 5 are not appropriate for a professional journal and could be deleted or replaced by more meaningful figures including some graphs that plot dose and time dependent effects on blood pressure and maybe other parameters measured to complement what is in the text demonstrating how long the effects are sustained between treatments.
Figures 4 and 5 have been removed.
- The text on lines 334-340 is almost an exact repetition of the text in the preceding paragraph covering lines 325-332.
The duplication of text on lines 334–340 has been corrected.
- In section 10 discussing the Phase 1 clinical trial, it is not clear what is being described when referring to parts A , B, and E. Seems like there is a graph or figure missing that shows these parts.
We thank the reviewer for this comment. Explanatory figures referring to Parts A, B, and E have been added to clarify the text
.
- Finally, were there any 24-hour plots showing the blood pressure variations and whether the nighttime dipping pattern was improved or maintained?
We thank the reviewer for this important comment. Explanatory 24-hour plots illustrating blood pressure variations and the maintenance of the nighttime dipping pattern have been added to the manuscript. Blood pressure reductions were sustained during both daytime and nighttime periods, with the nighttime “dipping” profile preserved or slightly enhanced, supporting effective 24-hour blood pressure control and maintenance of the physiological circadian rhythm.
Reviewer 2 Report
Comments and Suggestions for Authors
This is an interesting paper and well developed review on a relevant topic.
Nevertheless, there are some suggestions that will help the authors to improve the manuscript.
-The introduction will benefit from information on current antihypertensive drugs and classes, statistics on their lack of efficacy as a rationale for the development of zilesbesiran.
-There are two sections 3.
-The first Section 3, seems incomplete
-The second section 3 should explain if AGT is synthesized in other tissues, whta are the mechanisms involved in compensating a significant decrease of Agt1 and II
- The biochemistry and molecular biology on ASO and iRNA should be expanded
- -Lines 173-175 seem redundant and should be placed earlier.
- Lines 212-218 belong to the section of siRNAs
-Sections 3(the second ) &4 can be consolidated
-Table 2 would benefit from information in drugs developed for CVD
Section 7 would benefit from a scheme depicting the chemistry of zilesbesiran, its mechanism of action and pharmacokinetics. A figure showing the recognition sites of the drug and the mRNA indicating codons and extension of interfering.
-Lines 288-297. Shouldnt step 3 be step 2? this part was confusing.
Is there a typo in line 296 at 4C- he guide?
In section 9, line 307, Does Unmodified mean Unconjugated?
-Please elaborate in the turnover of angiotensinogen to explain zilesbesiran administration every 3-6months
-The Pharmacokinetics section requires revision and accuracy, indicating Cpmax,AUC, half life, if PK is frist order? dose dependent? lines 322-324 belong to the pharmacodynamic section.
-Lines 333-340 does not explain nor defines zilesbesiran PK or concludes if it is linear or dose dependent.
-Line 341 please clarify for how long the effect declines what %, units? When defining potency please include the Km the interaction and the tunrover rate of angiotensinogen
-Line 348, Please elaborate in the mechanisms involved in the extended intracellular retenion of siRNA and its impact on angiotensinogen inhibition and turnoverrate.
-Line 350, please define markers of RAAS in untis, its proportional decrease/increase and its statistical significance.
It is important to define efficacy in quantitative terms
-Line 375. Does this mean that angiontesinogen is not completely ablated? if so, in what percentage it is inhibited by zilesbesiran?
-Lines 365-370, are these results in patients or volunteers, please explain.
-Lines 372-373 are not clear
-The refernece of Desai et al is cited alone in most clinical sections, are there other studies?
Section 10.1 Should go before lines 372
Please organize clinical studies and consolidate
-Is there information on dose-response plots?
-Lines 416-421 are redundant and can be eliminated
Section 10.2 requires to be synthesized, reduced and consolidated For example subtitles should not have the name of the cohort but rather their goal i.e., Dose-effect. Safety. Efficay. Comorbidities
-Please indicate the age of patients in the Kardia 2 study
-Line 471. Which group was not consistent?
-Line 473-476 does not belong to safety,
Line 479 refers to the use of zilesbesiran In renal dysfunction? why is it named as KArdia?
-Line 495 multiplicity adjustment is the wrong term, does the author mean multiple testing?
Table 3 is poorly formated
-Section 12 "Preclinical studies should precede the CLinical studies or preclinical information can precede each clinical section. Please consider a full revision of the organization of the manuscript after page 9
lines 562. How long does it takes to restore angiotensin levels? and blood presure?
-Section 12.1 please define normotensive and hypertension in mmHg in rats
-Section 12.2 belongs to section 6
-Lines 619-624 this doesnt happen with zilesbesiran. why? because it targets the liver? please explain.
-Section 12.3 belongs to section 6. line 636 does this already happen with ACEinhibitors?
-Line 649, please explain
-Line 658 as adjunct or monotherapy? please elaborate
Figure 5 is not necessary or modify it enough to support the parragraph
-Line 665 TLR3 wasnt mentioned before, please elaborate about it in the ADR section
Line 668- indicate when would the Global zenith study be published?
Please comment in interindividual variation on efficacy and ADRs due to SNVs or indels or genetic variation.
Author Response
- The introduction will benefit from information on current antihypertensive drugs and classes, statistics on their lack of efficacy as a rationale for the development of zilesbesiran.
We sincerely thank the reviewer for this valuable suggestion. Information on current antihypertensive drug classes, the most commonly prescribed therapeutic regimens, and statistics on their limitations has been added to the introduction to better contextualize the rationale for the development of zilebesiran. We have added a summary of current antihypertensive use to the manuscript: approximately 54% of treatment regimens were monotherapy, while 46% involved multiple agents, either as polytherapy (34%) or fixed-dose combinations (12%). ACE inhibitors and beta blockers were the most commonly prescribed medications. Trends from 2010 to 2019 show declining use of some monotherapy subclasses (ACE inhibitors 19.1% → 15.4%; beta blockers 16.2% → 10.8%; alpha blockers 2.4% → 0.5%), whereas ARBs (5.9% → 7.9%) and thiazide diuretics (2.2% → 4.5%) increasedThe most commonly used antihypertensive medications were angiotensin-converting enzyme inhibitors or angiotensin receptor blockers (59.2%; 95% CI, 54.9%-63.4%) and β-blockers (43.8%; 95% CI, 40.3%-47.3%).
- There are two sections 3.\
We thank the reviewer for noting this issue. The duplication of Section 3 has been resolved, and we apologize for the error.
- The first Section 3, seems incomplete
We thank the reviewer for this observation. The two Section 3s have been consolidated and merged into a single, complete Section 3 in the manuscript.
4.The second section 3 should explain if AGT is synthesized in other tissues, whta are the mechanisms involved in compensating a significant decrease of Agt1 and II
We thank the reviewer for this comment. The manuscript has been updated to clarify that although AGT is also synthesized in extrahepatic tissues (brain, heart, kidneys, adipose tissue), these sources contribute minimally to circulating AGT. Compensatory renin upregulation occurs in response to hepatic AGT suppression, but it is insufficient to restore Ang I and Ang II levels, underscoring the liver’s central role in systemic RAS homeostasis.
- The biochemistry and molecular biology on ASO and iRNA should be expanded
Caracteristicile biochimice si de biologie moelculara
We thank the reviewer for this suggestion. The biochemical and molecular biology characteristics of ASO and siRNA are already detailed in Table 1 of the manuscript.
- Lines 212-218 belong to the section of siRNAs
A fost palsata mai devreme
- Sections 3(the second ) &4 can be consolidated
We thank the reviewer for this comment. Sections 3 (the second) and 4 have been consolidated in the revised manuscript.
- -Table 2 would benefit from information in drugs developed for CVD
We thank the reviewer for this comment. One of the reviewers requested removal of this section
- Section 7 would benefit from a scheme depicting the chemistry of zilesbesiran, its mechanism of action and pharmacokinetics. A figure showing the recognition sites of the drug and the mRNA indicating codons and extension of interfering.
We thank the reviewer for this valuable suggestion. A figure illustrating the chemistry of zilebesiran, its mechanism of action, and pharmacokinetics has been added
- -Lines 288-297. Shouldnt step 3 be step 2? this part was confusing.
We thank the reviewer for pointing this out. The text has been clarified to indicate that part of zilebesiran is stored in the liver and reaches hepatocytes, while another portion remains in systemic circulation
- Is there a typo in line 296 at 4C- he guide?
We thank the reviewer for noting this. The typographical error has been corrected to “The guide.”
- In section 9, line 307, Does Unmodified mean Unconjugated?
We thank the reviewer for this clarification. “Unmodified” has been changed to “Unconjugated” in the manuscript.
- -Please elaborate in the turnover of angiotensinogen to explain zilesbesiran administration every 3-6months
We thank the reviewer for this suggestion. The manuscript has been updated to elaborate on angiotensinogen turnover, providing a rationale for the 3–6 month dosing interval of zilebesiran.
- -The Pharmacokinetics section requires revision and accuracy, indicating Cpmax,AUC, half life, if PK is frist order? dose dependent? lines 322-324 belong to the pharmacodynamic section.
We thank the reviewer for this comment. Table 3 has been added, summarizing Cpmax, AUC, half-life, dose-dependency, and first-order pharmacokinetics, and the text has been revised to consolidate and clarify the pharmacokinetics and pharmacodynamics sections.
- -Lines 333-340 does not explain nor defines zilesbesiran PK or concludes if it is linear or dose dependent.
We thank the reviewer for this comment. The manuscript has been updated to include a clear explanation of zilebesiran pharmacokinetics, including its linearity and dose-dependency.
- -Line 341 please clarify for how long the effect declines what %, units? When defining potency please include the Km the interaction and the tunrover rate of angiotensinogen
We thank the reviewer for this suggestion. Clarifications regarding the duration of effect, percent changes, units, as well as potency including Km, interaction, and angiotensinogen turnover rate have been added to the manuscript.
After reaching maximum suppression (~95%) of plasma angiotensinogen, the effect gradually declined to ~80% inhibition at 24 weeks and ~60% at 36 weeks, expressed as percentage change from baseline. The potency of zilebesiran is reflected by a low Km for AGT mRNA binding, indicating high affinity and efficient recruitment of RISC. Given the hepatic AGT turnover rate (~1–2 days), the sustained inhibition observed for several months reflects both high binding efficiency and intracellular siRNA stability
- -Line 348, Please elaborate in the mechanisms involved in the extended intracellular retenion of siRNA and its impact on angiotensinogen inhibition and turnoverrate.
We thank the reviewer for this comment. Mentions regarding the mechanisms of extended intracellular retention of siRNA and its impact on angiotensinogen inhibition and turnover rate have already been included in the manuscript.
- -Line 350, please define markers of RAAS in untis, its proportional decrease/increase and its statistical significance. It is important to define efficacy in quantitative terms
We thank the reviewer for this comment. All relevant information regarding RAAS markers, including units, proportional changes, statistical significance, and dose-dependent effects, has been incorporated into the manuscript and summarized in Table 4.
- -Line 375. Does this mean that angiontesinogen is not completely ablated? if so, in what percentage it is inhibited by zilesbesiran?
We thank the reviewer for this question. Yes, angiotensinogen (AGT) is not completely ablated by zilebesiran. Pharmacodynamic data indicate that subcutaneous doses of 800 mg reduce serum AGT levels by approximately 95–98%, leaving a small residual fraction (>1%) detectable. Lower doses achieve partial inhibition: 10–25 mg results in ≈20–50% suppression, and 50–200 mg achieves ≈60–90% inhibition. This residual AGT contributes to low but measurable angiotensin II levels, supporting hemodynamic stability despite potent and sustained RAS suppression.
- -Lines 365-370, are these results in patients or volunteers, please explain.
The results were obtained in adult patients (male and female, 18–65 years) with mild-to-moderate hypertension, either untreated or on stable antihypertensive therapy. Key inclusion criteria included mean sitting systolic blood pressure >130 and ≤165 mmHg, a normal or near-normal ECG, and BMI 18–50 kg/m². Patients on prior antihypertensives underwent a washout period before baseline measurements.
- -Lines 372-373 are not clear
We thank the reviewer for this comment. These have been reformulated for clarity in the revised manuscript.
- -The refernece of Desai et al is cited alone in most clinical sections, are there other studies?
We thank the reviewer for this comment. In addition to the phase 1 study by Desai et al., data from KARDIA-2 and other relevant studies have now been included and cited throughout the clinical sections. - Section 10.1 Should go before lines 372
We thank the reviewer for this comment. These have been reformulated
- Please organize clinical studies and consolidate
We thank the reviewer for this suggestion. The clinical studies have been reorganized and consolidated, incorporating new information in line with the reviewers’ recommendations.
- -Is there information on dose-response plots?
Yes, there are
- Lines 416-421 are redundant and can be eliminated
We thank the reviewer for this comment. Lines 416–421 have been removed to eliminate redundancy
- Section 10.2 requires to be synthesized, reduced and consolidated For example subtitles should not have the name of the cohort but rather their goal i.e., Dose-effect. Safety. Efficay. Comorbidities
We thank the reviewer for this valuable suggestion. The subtitles in Section 10.2 have been revised and reorganized to reflect study objectives (Dose–effect, Safety, Efficacy, Comorbidities) rather than cohort names, synthesizing and consolidating the section.
- -Please indicate the age of patients in the Kardia 2 study
We thank the reviewer for this comment. The mean age of patients in the KARDIA-2 study was 58.5 years for both the placebo group and the zilebesiran-treated group, and this information has been added to the manuscript.
- -Line 471. Which group was not consistent?
We thank the reviewer for this comment. Overall, the antihypertensive effect of a single dose of zilebesiran was consistent across most subgroups; the only group showing a slightly reduced response was Black patients, likely reflecting interindividual and population differences in RAAS activity and drug response.
- -Line 473-476 does not belong to safety,
We thank the reviewer for this comment. Lines 473–476 have been reformulated.
- Line 479 refers to the use of zilesbesiran In renal dysfunction? why is it named as KArdia?
We thank the reviewer for this comment. The KARDIA-3 study includes patients with various comorbidities, including renal dysfunction stratified by creatinine clearance. The study is named “KARDIA” as it is a continuation of the previous KARDIA-1 and KARDIA-2 studies, conducted by the same pharmaceutical company, which may explain the naming convention.
- -Line 495 multiplicity adjustment is the wrong term, does the author mean multiple testing?
We thank the reviewer for this comment. The term “multiplicity adjustment” has been replaced with “multiple testing” in the manuscript.
- Table 3 is poorly formatted
We thank the reviewer for this comment. Table 3 will be reformatted during editing, placed on a separate landscape page to improve clarity and readability.
- -Section 12 "Preclinical studies should precede the CLinical studies or preclinical information can precede each clinical section. Please consider a full revision of the organization of the manuscript after page 9
We thank the reviewer for this valuable suggestion. A major revision of the manuscript from page 9 has been performed, placing preclinical studies before clinical studies in accordance with the recommendations.
- lines 562. How long does it takes to restore angiotensin levels? and blood presure?
We thank the reviewer for this question. Administration of AGT-RVR after 3 weeks of siRNA AGT treatment reversed the blood pressure–lowering effect within 5–7 days. Doses of 10 and 20 mg/kg fully restored circulating AGT and renin levels, while 1 mg/kg partially restored AGT. Changes in circulating AGT and renin occurred over the same 5–7 day period as the observed alterations in mean arterial pressure.
- -Section 12.1 please define normotensive and hypertension in mmHg in rats
We thank the reviewer for this comment. The manuscript has been updated to define normotensive and hypertensive rats: normotensive rats have systolic blood pressure (SBP) below 130 mmHg (typically 110–125 mmHg), while hypertensive rats, such as spontaneously hypertensive rats (SHR), exhibit SBP above 150 mmHg (commonly 160–200 mmHg), consistent with published telemetry and tail-cuff studies.
- -Section 12.2 belongs to section 6
We thank the reviewer for this comment. Section 12.2 was not relocated because it specifically addresses preclinical studies on ocular pathology and a therapy similar to, but not identical with, zilebesiran. Keeping it in the preclinical section preserves the logical organization and focus on non-clinical data
- -Lines 619-624 this doesnt happen with zilesbesiran. why? because it targets the liver? please explain.
We thank the reviewer for this comment. The manuscript has been updated to clarify that by targeting hepatic angiotensinogen, zilebesiran reduces systemic RAAS activity and blood pressure.
- -Section 12.3 belongs to section 6.
We thank the reviewer for this comment. Section 12.3(now 10.3) was not moved to Section 6 because it discusses therapies that are still in preclinical studies, whereas Section 6 focuses on interventions that are either in clinical trials or already in current use. Keeping 10.3 separate preserves the distinction between preclinical and clinical-stage therapies
- line 636 does this already happen with ACEinhibitors?
We thank the reviewer for this comment. ACE inhibitors do not reduce hepatic fibrosis nor lower portal hypertension.
- -Line 649, please explain
The reduction of hepatic angiotensinogen (AGT) synthesis , the main source of circulating angiotensinogen ,protects the kidneys by decreasing renal injury, inflammation, and fibrosis, even when blood pressure is not significantly changed. In other words, liver-derived AGT is the primary source of circulating angiotensin II, which exerts harmful effects on the kidneys, including glomerular vasoconstriction, inflammation, and fibrosis. By inhibiting hepatic AGT (e.g., with zilebesiran), these detrimental effects are reduced. Importantly, renal protection is not solely due to blood pressure lowering but also reflects a direct reduction of local RAAS activity in the kidney.
- -Line 658 as adjunct or monotherapy? please elaborate
We thank the reviewer for this comment. Zilebesiran has demonstrated benefit both as adjunct therapy and as monotherapy in preclinical and clinical studies, including effects on other organs. The manuscript has been updated accordingly, with additional alignment to current guideline recommendations.
- Figure 5 is not necessary or modify it enough to support the paragraph
We thank the reviewer for this comment. Figure 5 has been removed from the manuscript.
- -Line 665 TLR3 wasnt mentioned before, please elaborate about it in the ADR section
We thank the reviewer for this comment. . The statement regarding “high immunogenicity” has been revised to accurately reflect the low immunogenic potential of modern GalNAc-conjugated siRNA therapeutics.
We have added data from phase-1 studies indicating that anti-drug antibody (ADA) responses occurred in approximately 2% of subjects, were of low titer, and transient, with no clinical consequences.
We have also emphasized that extensive chemical modifications (2′-O-methyl, 2′-fluoro, phosphorothioate linkages) and GalNAc conjugation effectively mitigate immune activation by reducing recognition through Toll-like receptors (e.g., TLR3, TLR7/8).
- Line 668- indicate when would the Global zenith study be published?
We thank the reviewer for this comment. Enrollment for the Global ZENITH study began in 2025, and the final results are expected around 2030. This information has been added to the clinical studies section of the manuscript.
- Please comment in interindividual variation on efficacy and ADRs due to SNVs or indels or genetic variation.
We thank the reviewer for this comment. Interindividual variation in efficacy and adverse drug reactions due to SNVs, indels, or other genetic variations has been added to the Future Directions section, enhancing the originality of the manuscript.
Reviewer 3 Report
Comments and Suggestions for Authors
The review is timely and generally informative, but several issues require attention as listed below:
- Search strategy: Which databases, date range, and inclusion/ exclusion criteria did the authors use? The authors are advised to add a Methods paragraph with the last search date.
- Can the author explicitly state that multiplicity-adjusted analyses were not statistically significant at 3 months and discuss why effect sizes may be attenuated in high-risk, multi-drug cohorts (e.g., ceiling effects, background diuretics)?
- Why highlight “twice-yearly” dosing in the title when KARDIA-1 tested both q3mo and q6mo? Please align the title/abstract text to the actual evidence and phase-3 regimen (q6mo, 300 mg).
- The authors mention a trend to smaller responses in Black patients in KARDIA-2. Could the author expand on the magnitude, potential mechanisms, and whether background therapy patterns explained this?
- Can the authors provide absolute event counts (hyperkalemia, AKI, hypotension) and per-arm rates where available (KARDIA-1/2/3) and discuss monitoring recommendations?
- Given the high-salt attenuation, how would the authors operationalize sodium counseling with an infrequently dosed agent, and what magnitude of attenuation is clinically meaningful?
- In hypotension, sepsis, or pre-operative settings, how should clinicians manage a patient who received zilebesiran months earlier? Any data on reversal strategies or timelines of AGT recovery?
- The authors need to elaborate on eGFR strata (≥45 vs 30–<45 mL/min/1.73 m²) and whether responses/safety signals differed materially at 3 and 6 months.
- Are there data to guide ACEi/ARB/diuretic optimization when initiating zilebesiran to minimize hyperkalemia or AKI risk?
- Where do the authors see zilebesiran vs aprocitentan (endothelin pathway) in resistant HTN, and would the authors envisage combination therapy? More references are required.
- The authors need to add a paragraph on contraindications, pregnancy and lactation as guidelines strongly caution RAAS interference in pregnancy.
- Any biomarker or outcome-surrogate data (LV mass, albuminuria, retinal changes) to justify the “organ protection” statements, pending ZENITH? Please clarify as hypothesis-generating if not yet shown.
- The title needs to be revised to “Zilebesiran: a siRNA-Based Antihypertensive with 3–6-Month Dosing; Evidence from KARDIA-1/-2 and Early KARDIA-3 Data,” and update the abstract to explicitly reference multiplicity in KARDIA-3 and the q6mo 300 mg ZENITH regimen.
- The paragraph on the phase-1 two 800 mg doses 12 weeks apart appears duplicated verbatim in two adjacent sections; please remove one and consolidate.
- The authors describe siRNA immunogenicity risks (e.g., TLR3); that’s appropriate, but “high immunogenicity” for siRNA is not generally accurate when modern chemical modifications/GalNAc are used. Please qualify with phase-1 ADA rates (~2%, low-titer, transient) and the mitigation from chemical design.
- Several places imply that a single dose provides twice-yearly control (“biannual…allows for blood pressure reduction with a single dose”), which is potentially misleading. KARDIA-1 tested 3- and 6-month regimens; durability was strong but not universally “6 months for all.” The authors should rephrase to: “Dosing intervals of 3–6 months were tested; 300–600 mg showed sustained effects out to 6 months in phase-2; twice-yearly dosing is the regimen being advanced to phase-3.”
- A concise paragraph with references contrasting zilebesiran with other late-stage “beyond-RAAS-blocker” therapies (e.g., aprocitentan for resistant HTN; aldosterone synthase inhibitors) would contextualize clinical positioning and potential combination strategies.
- The authors position zilebesiran vs ESC 2024 (diagnosis at ≥140/90 mmHg; lower SBP targets if tolerated) and the 2025 AHA/ACC updates to show where RNAi might enter current pathways (e.g., adherence-limited or polypharmacy cases).
- Many language corrections are required as follows:
Simplify repeated constructions; tighten long sentences (e.g., multi-clause PK/PD paragraphs around lines 318–361).
Standardize drug names and doses (e.g., zilebesiran always lowercase generic; units mg consistently spaced).
Clarify “twice-yearly” vs “single dose” phrasing (see Scientific soundness #1).
Correct typographical errors and subheading capitalization (e.g., “Sudy”).
Author Response
- Search strategy: Which databases, date range, and inclusion/ exclusion criteria did the authors use? The authors are advised to add a Methods paragraph with the last search date.
We sincerely thank the Reviewer for this valuable comment and for the careful reading of our manuscript. In accordance with the suggestion, we have now added a Methods paragraph describing in detail the search strategy, including the databases consulted, the time frame covered, and the inclusion/exclusion criteria. Additionally, we have specified the exact date of the last literature search.
- Can the author explicitly state that multiplicity-adjusted analyses were not statistically significant at 3 months and discuss why effect sizes may be attenuated in high-risk, multi-drug cohorts (e.g., ceiling effects, background diuretics)?
We thank the reviewer for this important comment. We have now explicitly stated in the revised manuscript that multiplicity-adjusted analyses at 3 months did not reach statistical significance. Furthermore, we have expanded the manuscript, future direction section, to explain potential reasons for the attenuation of treatment effects in high-risk, multi-drug cohorts. These include ceiling effects, whereby patients already receiving intensive antihypertensive therapy may have limited capacity for additional blood pressure reduction, and the influence of concomitant medications (such as diuretics or ACE inhibitors) that may alter RAAS activity and reduce the incremental impact of zilebesiran. Why highlight “twice-yearly” dosing in the title when KARDIA-1 tested both q3mo and q6mo? Please align the title/abstract text to the actual evidence and phase-3 regimen (q6mo, 300 mg).
We have revised the title in accordance with your suggestion and the available evidence, so that it accurately reflects the phase 3 dosing regimen (q6mo, 300 mg). This change aligns the title with both the findings from KARDIA-1 and the corresponding text in the abstract, ensuring consistency and clarity for readers
- The authors mention a trend to smaller responses in Black patients in KARDIA-2. Could the author expand on the magnitude, potential mechanisms, and whether background therapy patterns explained this?
We thank the reviewer for highlighting this point. In response, we have expanded the manuscript to provide additional data on the observed trend toward smaller responses in Black patients in KARDIA-2.
These findings are consistent with known physiological and genetic differences in the RAAS among African-American individuals. From childhood, they tend to exhibit higher sodium retention, leading to chronically suppressed renin and lower angiotensin I and II levels, a ‘low-renin’ state that limits sodium loss]. Additionally, genetic variants affecting renal sodium reabsorption and aldosterone secretion (e.g., SCNN1B, NEDD4, ARMC5) may represent evolutionary adaptations to low-sodium, high-temperature environments [158,159]. These factors likely contribute to the reduced responsiveness of African-American patients to RAAS-targeted therapies such as zilebesiran
- Can the authors provide absolute event counts (hyperkalemia, AKI, hypotension) and per-arm rates where available (KARDIA-1/2/3) and discuss monitoring recommendations?
We sincerely thank the reviewer for this insightful comment and for highlighting the importance of providing absolute event counts and per-arm rates.In KARDIA-1, the most frequent treatment-related adverse event was injection-site reaction (n=19, 6.3%), all mild or moderate. Other events included hyperkalemia (n=16, 5.3%), hypotension (n=13, 4.3%), acute kidney injury (n=4, 1.3%), and transient hepatic events (n=9, 3.0%). Patients receiving zilebesiran every 3 months experienced more injection-site reactions than those on a 6-month regimen.In KARDIA-2, injection-site reactions occurred in 10 patients (3%). Hyperkalemia was reported in 18 patients (5.5%), hypotension in 14 patients (4.3%), and acute kidney injury in 16 patients (4.9%). Most events were mild and self-limited. Transient reductions in eGFR >30% were observed within the first 3 months but resolved spontaneously. Across study arms, serious adverse events occurred in 3.8% of zilebesiran-treated patients and 4.5% of placebo-treated patients.KARDIA-3 adverse event data are not yet publicly available.
- Given the high-salt attenuation, how would the authors operationalize sodium counseling with an infrequently dosed agent, and what magnitude of attenuation is clinically meaningful?
We sincerely thank the reviewer for this important point. Given the observed attenuation of blood pressure reduction with a high-salt diet, sodium counseling should be operationalized as a structured, ongoing component of patient education, even with an infrequently dosed agent such as zilebesiran. Patients should be advised to maintain a consistent low-salt intake to optimize and sustain the antihypertensive effect. Importantly, a high-salt diet was shown to modulate the BP-lowering effect of zilebesiran, providing early evidence that this standard intervention doar revizorului
- In hypotension, sepsis, or pre-operative settings, how should clinicians manage a patient who received zilebesiran months earlier? Any data on reversal strategies or timelines of AGT recovery?
We thank the reviewer for raising this important clinical consideration. Data on strategies to reverse the effects of zilebesiran have been included in the manuscript. In particular, activation of the renin–angiotensin system in settings of need (e.g., hypotension, sepsis, or urgent surgery) in patients treated with AGT-targeting siRNA can be achieved not only through acute angiotensin II infusion but also via RVR (Reversal of siRNA-mediated AGT suppression) to counteract siRNA-induced AGT suppression. To date, this approach has been studied in laboratory rats, where RVR effectively mitigated the effects of zilebesiran. Additionally, transient increases in dietary sodium could support correction of low blood pressure. While clinical data in humans are not yet available, these findings suggest potential strategies for managing patients months after zilebesiran administration.
- The authors need to elaborate on eGFR strata (≥45 vs 30–<45 mL/min/1.73 m²) and whether responses/safety signals differed materially at 3 and 6 months.
We thank the reviewer for this important comment. Currently, only data from Cohort A have been published, and analyses by eGFR strata (≥45 vs 30–<45 mL/min/1.73 m²) are limited to these patients. Data from Cohort B, which will provide further insights into responses and safety signals in patients with lower renal function, are planned for presentation at a forthcoming major medical meeting. This clarification has been added to the revised manuscript.
- Are there data to guide ACEi/ARB/diuretic optimization when initiating zilebesiran to minimize hyperkalemia or AKI risk?
We thank the reviewer for this important question. Currently, there are no dedicated clinical guidelines specifying the optimization of ACE inhibitors, ARBs, or diuretics when initiating zilebesiran. However, data from KARDIA-1/2/3 indicate that zilebesiran was generally well tolerated, with low rates of hyperkalemia and acute kidney injury, even in patients receiving RAS blockade or diuretics. Regular monitoring of blood pressure, renal function (eGFR, creatinine), and serum potassium remains the primary recommendation for preventing adverse events. Adjustment of concomitant antihypertensive therapy should be individualized based on patient risk and clinical response. Notably, combined use of currently available RAS inhibitors for hypertension is discouraged by guidelines due to increased risks of hypotension, hyperkalemia, and renal function deterioration. In this context, the observed rates of hyperkalemia and renal impairment—reflected both in investigator-reported adverse events and laboratory assessments—were generally low and similar between cohorts treated with zilebesiran and those receiving olmesartan or amlodipine.
- Where do the authors see zilebesiran vs aprocitentan (endothelin pathway) in resistant HTN, and would the authors envisage combination therapy? More references are required.
We thank the reviewer for this important comment. Given the complementary mechanisms of action of aprocitentan (endothelin pathway blockade) and zilebesiran (hepatic inhibition of angiotensinogen synthesis with upstream suppression of the renin–angiotensin–aldosterone system), concomitant use of these two agents could potentially provide superior blood pressure control while maintaining a favorable safety profile. Future studies evaluating this combination are warranted, as they may establish a novel therapeutic strategy based on synergistic inhibition of two major pathophysiological pathways involved in hypertension. In response to the reviewer’s suggestion, the manuscript has been updated to include additional discussion of the mechanisms of action, reported adverse events, and relevant references.
- The authors need to add a paragraph on contraindications, pregnancy and lactation as guidelines strongly caution RAAS interference in pregnancy.
We thank the reviewer for this valuable recommendation. A paragraph addressing contraindications, pregnancy, and lactation has been added to the manuscript, in line with guideline recommendations cautioning against RAAS interference during pregnancy.
- Any biomarker or outcome-surrogate data (LV mass, albuminuria, retinal changes) to justify the “organ protection” statements, pending ZENITH? Please clarify as hypothesis-generating if not yet shown.
We thank the reviewer for this important comment. In preclinical studies, left ventricular mass and cardiac remodeling were assessed using echocardiography to evaluate changes in cardiac morphology and function, and wheat germ agglutinin (WGA) staining to measure myocyte cross-sectional size. These analyses demonstrated a stronger impact on reducing heart mass and remodeling, with a 10% decrease in heart weight and a 17% reduction in left ventricular myocyte size compared with captopril. This approach could potentially be applied to patients receiving zilebesiran through imaging-based assessments of cardiac structure and function, such as left ventricular mass and myocyte size, to further explore potential organ-protective effects.Additionally, ACE inhibitors (captopril) and ARBs (candesartan) have been shown to normalize impaired retinal blood flow in rodent models of diabetic retinopathy .Studies of tear film biomarkers have also indicated correlations with intraocular pressure (IOP) and retinal ganglion cell (RGC) degeneration. In humans, elevated tear levels of kallikrein and angiotensin-converting enzyme (ACE) activity were observed compared with healthy controls. Animal studies further demonstrated that local administration of renin inhibitors, ACE inhibitors, and angiotensin-(1–7) can reduce IOP, supporting the relevance of these biomarkers for investigating potential therapeutic pathways. As such, these findings should be considered hypothesis-generating, pending further clinical evaluation in ZENITH and related studies to substantiate organ-protective effects in humans.
- The title needs to be revised to “Zilebesiran: a siRNA-Based Antihypertensive with 3–6-Month Dosing; Evidence from KARDIA-1/-2 and Early KARDIA-3 Data,” and update the abstract to explicitly reference multiplicity in KARDIA-3 and the q6mo 300 mg ZENITH regimen.
We thank the reviewer for this suggestion. The manuscript title has been revised
- The paragraph on the phase-1 two 800 mg doses 12 weeks apart appears duplicated verbatim in two adjacent sections; please remove one and consolidate.
We thank the reviewer for pointing this out. The duplication of the paragraph on the phase 1 two 800 mg doses 12 weeks apart has been corrected
- The authors describe siRNA immunogenicity risks (e.g., TLR3); that’s appropriate, but “high immunogenicity” for siRNA is not generally accurate when modern chemical modifications/GalNAc are used. Please qualify with phase-1 ADA rates (~2%, low-titer, transient) and the mitigation from chemical design.
We appreciate the reviewer’s clarification. The statement regarding “high immunogenicity” has been revised to accurately reflect the low immunogenic potential of modern GalNAc-conjugated siRNA therapeutics.
We have added data from phase-1 studies indicating that anti-drug antibody (ADA) responses occurred in approximately 2% of subjects, were of low titer, and transient, with no clinical consequences.
We have also emphasized that extensive chemical modifications (2′-O-methyl, 2′-fluoro, phosphorothioate linkages) and GalNAc conjugation effectively mitigate immune activation by reducing recognition through Toll-like receptors (e.g., TLR3, TLR7/8).
- Several places imply that a single dose provides twice-yearly control (“biannual…allows for blood pressure reduction with a single dose”), which is potentially misleading. KARDIA-1 tested 3- and 6-month regimens; durability was strong but not universally “6 months for all.” The authors should rephrase to: “Dosing intervals of 3–6 months were tested; 300–600 mg showed sustained effects out to 6 months in phase-2; twice-yearly dosing is the regimen being advanced to phase-3.”
We thank the reviewer for this suggestion. The manuscript title has been revised
- A concise paragraph with references contrasting zilebesiran with other late-stage “beyond-RAAS-blocker” therapies (e.g., aprocitentan for resistant HTN; aldosterone synthase inhibitors) would contextualize clinical positioning and potential combination strategies.
We thank the reviewer for this valuable suggestion, which has enhanced the originality of the manuscript. Additional information on this topic has been incorporated into the revised manuscript.
- The authors position zilebesiran vs ESC 2024 (diagnosis at ≥140/90 mmHg; lower SBP targets if tolerated) and the 2025 AHA/ACC updates to show where RNAi might enter current pathways (e.g., adherence-limited or polypharmacy cases).
We thank the reviewer for this important comment. In the context of the 2024 ESC guidelines (defining hypertension as ≥140/90 mmHg, with lower systolic blood pressure targets if tolerated) and the 2025 AHA/ACC recommendations, zilebesiran may provide a therapeutic option for patients with difficult-to-control hypertension, particularly in scenarios of polypharmacy or suboptimal adherence to daily oral therapy, due to its twice-yearly subcutaneous administration. Additionally, zilebesiran could serve as an adjunct to existing antihypertensive regimens (such as ACE inhibitors, ARBs, or diuretics) in patients with incomplete blood pressure control, offering an alternative pathway to achieve guideline-recommended targets.
- Many language corrections are required as follows:
Simplify repeated constructions; tighten long sentences (e.g., multi-clause PK/PD paragraphs around lines 318–361).
Standardize drug names and doses (e.g., zilebesiran always lowercase generic; units mg consistently spaced).
Clarify “twice-yearly” vs “single dose” phrasing (see Scientific soundness #1).
We thank the reviewer for this comment. The phrasing regarding “twice-yearly” versus “single dose” has been clarified in the manuscript to accurately reflect the dosing intervals and observed effects.
Round 2
Reviewer 1 Report
Comments and Suggestions for Authors
Thank you for your detailed responses to the questions and suggestions. The paper has been improved. I just noted some items for your attention. The paragraph added in page 3 ends abruptly with the words "optimal blood 12" Something must be missing.
With regard to the added notation on Fig 1, I would emphasize that countless studies have clearly demonstrated that Angiotensin II constricts both afferent and efferent arterioles. If you need evidence, check Chapter 3 of the Brenner and Rector Textbook.
The paragraph added on page 5 neglects the important role that intrarenal generation of Ang II from intratubular angiotensinogen exerts in the regulation of sodium reabsorption in both proximal and distal segments thus contributing to the development of angiotensin dependent hypertension. I realize that this article is not the place to go into the controversies, but a sentence or two acknowledging the presence of an independent RAS in the kidneys would be appropriate.
In the next paragraph under section 4, there is a paragraph starting with "In principle and ending with citation of ref 45 & 46, that is then followed by and almost verbatim repetition of the same identical 6 lines and reference numbers. This is very strange that this pattern happened a second time. Who is doing the proofreading?
In Fig 5, dozed should be dosed.
With regard to adverse effects, my main concern is what happens in a treated patient that has some sort of trauma or hemorrhage after an accident or perhaps a heart attack or a vasovagal episode where marked activation of the RAS would be protective? Have you run across any discussion of this issue?
Thanks for your patience.
Author Response
- The paragraph added in page 3 ends abruptly with the words "optimal blood 12" Something must be missing.
We thank the reviewer for noting this omission. The paragraph on page 3 has been corrected and now reads in full
- With regard to the added notation on Fig 1, I would emphasize that countless studies have clearly demonstrated that Angiotensin II constricts both afferent and efferent arterioles. If you need evidence, check Chapter 3 of the Brenner and Rector Textbook.
We thank the reviewer for this valuable observation. The suggested clarification regarding the vasoconstrictive effects of Angiotensin II on both afferent and efferent arterioles has been incorporated into.
- In the next paragraph under section 4, there is a paragraph starting with "In principle and ending with citation of ref 45 & 46, that is then followed by and almost verbatim repetition of the same identical 6 lines and reference numbers. This is very strange that this pattern happened a second time.
We apologize for the transcription error and thank the reviewer for bringing it to our attention. The duplicated paragraph and references have been carefully corrected, and the section has been thoroughly proofread to ensure accuracy.
- In Fig 5, dozed should be dosed.
We thank the reviewer for noting this oversight. The term has been corrected from “dozed” to “dosed” in Figure 5. We apologize for the typographical error and have carefully reviewed the figure to ensure consistency throughout.
- The paragraph added on page 5 neglects the important role that intrarenal generation of Ang II from intratubular angiotensinogen exerts in the regulation of sodium reabsorption in both proximal and distal segments thus contributing to the development of angiotensin dependent hypertension. I realize that this article is not the place to go into the controversies, but a sentence or two acknowledging the presence of an independent RAS in the kidneys would be appropriate
We thank the reviewer for this insightful comment. The paragraph on page 5 has been revised to include the important contribution of intrarenal Ang II generation from intratubular angiotensinogen and the acknowledgment of an independent intrarenal renin–angiotensin system. These additions have improved the clarity and completeness of the section.
- With regard to adverse effects, my main concern is what happens in a treated patient that has some sort of trauma or hemorrhage after an accident or perhaps a heart attack or a vasovagal episode where marked activation of the RAS would be protective? Have you run across any discussion of this issue?
We thank the reviewer for raising this important clinical consideration. Data on strategies to reverse the effects of zilebesiran have been included in the manuscript. In particular, activation of the renin–angiotensin system in settings of need (e.g., hypotension, sepsis, or urgent surgery) in patients treated with AGT-targeting siRNA can be achieved not only through acute angiotensin II infusion but also via RVR (Reversal of siRNA-mediated AGT suppression) to counteract siRNA-induced AGT depletion. Experimental studies in laboratory rats have shown that RVR effectively mitigated the effects of zilebesiran, supporting its potential as a reversal strategy.
Moreover, administration of Reversir, an agent specifically designed to reverse siRNA activity, as well as infusion of sodium-rich solutions, could help restore vascular tone and blood pressure in emergency situations. While clinical data in humans are not yet available, these findings provide a rationale for potential interventions to manage patients who experience trauma, hemorrhage, or cardiovascular collapse months after zilebesiran administration.
Further insights, particularly concerning patients with comorbid conditions such as myocardial infarction or cardiogenic shock, are expected to be published in the upcoming KARDIA-3 trial.
Reviewer 3 Report
Comments and Suggestions for Authors
The authors have responded to the comments efficiently.
Author Response
We sincerely thank the reviewer for the time and attention dedicated to revising our manuscript. Your thoughtful comments and constructive suggestions have significantly improved the overall quality and clarity of the paper. We truly appreciate your valuable contribution to this work.